# Score-based Conditional Generation with Fewer Labeled Data by Self-calibrating Classifier Guidance

## Abstract

Score-based generative models (SGMs) are a popular family of deep generative models that achieve leading image generation quality. Early studies extend SGMs to tackle class-conditional generation by coupling an unconditional SGM with the guidance of a trained classifier. Nevertheless, such classifier-guided SGMs do not always achieve accurate conditional generation, especially when trained with fewer labeled data. We argue that the problem is rooted in the classifier's tendency to overfit without coordinating with the underlying unconditional distribution. We propose improving classifier-guided SGMs by letting the classifier regularize itself to respect the unconditional distribution. Our key idea is to use principles from energy-based models to convert the classifier as another view of the unconditional SGM. Then, existing loss for the unconditional SGM can be leveraged to achieve regularization by calibrating the classifier's internal unconditional scores. The regularization scheme can be applied to not only the labeled data but also unlabeled ones to further improve the classifier. Empirical results show that the proposed approach significantly improves conditional generation quality across various percentages of fewer labeled data. The results confirm the potential of the proposed approach for generative modeling with limited labeled data.

## 1 Introduction

Score-based generative models (SGMs) capture the underlying data distribution by learning the gradient function of the log-likelihood on data, also known as the score function. SGMs, when coupled with a diffusion process that gradually converts noise to data, can often synthesize higher-quality images than other popular alternatives, such as generative adversarial networks (Song et al., 2021; Dhariwal & Nichol, 2021). The community's research dedication on SGMs demonstrates promising performance in image generation (Song et al., 2021) and other fields such as audio synthesis (Kong et al., 2021; Jeong et al., 2021; Huang et al., 2022) and natural language generation (Li et al., 2022). Many such successful SGMs focus on unconditional generation, which models the distribution without considering other variables (Song & Ermon, 2019; Ho et al., 2020; Song et al., 2021). When seeking to generate images controllably from a particular class, it is necessary to model the conditional distribution concerning another variable. Such *conditional* SGMs (Song et al., 2021; Dhariwal & Nichol, 2021; Chao et al., 2022) will be the focus of this paper.

There are two major families of conditional SGMs. Classifier-free SGMs (CFSGMs) adopt specific conditional network architectures and losses (Dhariwal & Nichol, 2021; Ho & Salimans, 2021). Such SGMs are known to generate high-fidelity images when all data are labeled. Nevertheless, our findings indicate that their performance drops significantly as the proportion of labeled data decreases. This makes them less preferable in the semi-supervised setting with fewer labeled data, which is a realistic scenario when obtaining class labels takes significant time and costs. Classifier-guided SGMs (CGSGMs) form another family of conditional SGMs (Song et al., 2021; Dhariwal & Nichol, 2021) based on decomposing the conditional score into the unconditional score plus the gradient of an auxiliary classifier. A vanilla CGSGM can then be constructed by learning a classifier in parallel to training an unconditional SGM with the popular denoising score matching (DSM; Vincent, 2011) technique. The additional classifier can control the trade-off between generation diversity and fidelity better (Dhariwal & Nichol, 2021). Furthermore, because the unconditional SGM can

be trained with both labeled and unlabeled data in principle, CGSGMs emerge with more potential than CFSGMs for the semi-supervised setting with fewer labeled data.

The quality of the classifier gradients is critical for CGS-GMs. If the classifier overfits (Lee et al., 2018; Müller et al., 2019; Mukhoti et al., 2020; Grathwohl et al., 2020) and predicts highly inaccurate gradients, the resulting conditional scores may be unreliable, which lowers the generation quality even if the reliable unconditional scores can ensure decent generation fidelity. Although there are general regularization techniques (Zhang et al., 2019; Müller et al., 2019; Hoffman et al., 2019) that mitigate overfitting, their specific benefits for CGSGMs have not been fully studied except for a few cases (Kawar et al., 2022). In fact, we find that those techniques are often not

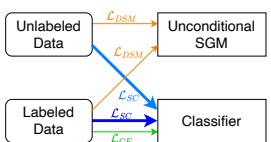

Figure 1: Illustration of proposed approach. A vanilla CGSGM takes the orange (DSM loss) and green (cross-entropy loss) arrows. The proposed CGSGM-SC additionally considers the two blue arrows representing the proposed self-calibration loss on both labeled and unlabeled data.

aligned with the unconditional SGM's view of the underlying distribution and offer limited benefits for improving CGSGMs. One pioneering enhancement of CGSGM on distribution alignment, denoising likelihood score matching (CG-DLSM; Chao et al., 2022), calibrates the classifier with a regularization loss that aligns the classifier's outputs to the unconditional SGM's outputs on labeled data. CG-DLSM addresses the alignment issue with the *external* help of the unconditional SGM and achieves state-of-the-art performance within the CGSGM family. However, DLSM is only designed for labeled data and is not applicable to unlabeled data.

In this work, we design a regularization term that calibrates the classifier *internally*, without relying on the unconditional SGM. Such an internal regularization has been previously achieved by the joint energy-based model (JEM; Grathwohl et al., 2020), which interprets classifiers as energy-based models. The interpretation allows JEM to define an auxiliary loss term that respects the underlying distribution and can unlock the generation capability of classifiers when using MCMC sampling. Nevertheless, extending JEM as CGSGM is non-trivial, as the sampling process is time-consuming and results in unstable loss values when coupled with the diffusion process of CGSGM. We thus take inspiration from JEM to derive a novel CGSGM regularizer instead of extending JEM directly.

Our design broadens the JEM interpretation of classifiers to be unconditional SGMs. Then, a stable and efficient self-calibration (SC) loss (as illustrated with $\mathcal{L}_{\text{SC}}$ in Fig. 1) can be computed from the classifier *internally* for regularization. The SC loss inherits a sound theoretical guarantee from the DSM technique for training unconditional SGMs. Our proposed CGSGM-SC approach, as shown in Fig. 1, allows separate training of the unconditional SGM and the classifier. The approach applies the SC loss on both labeled and *unlabeled* data, resulting in immediate advantages in the semi-supervised setting with fewer labeled data.

Following earlier studies on CGSGMs (Chao et al., 2022), we visually study the effectiveness of CGSGM-SC on a synthesized data set. The results reveal that the CGSGM-SC leads to more accurate classifier gradients than vanilla CGSGM, thus enhancing the estimation of conditional scores. We further conduct thorough experiments on CIFAR-10 and CIFAR-100 datasets to validate the advantages of CGSGM-SC. The results confirm that CGSGM-SC is superior to the vanilla CGSGM and the state-of-the-art CGSGM-DLSM approach. Furthermore, in an extreme setting for which only 5% of the data is labeled, CGSGM-SC, which more effectively utilizes unlabeled data, is significantly better than all CGSGMs and CFSGMs. This confirms the potential of CGSGM-SC in scenarios where labeled data are costly to obtain. We summarize the contributions of this paper as:

- We proposed to further reinterpret classifiers as SGMs for regularization.
- We discovered the potential of CGSGMs in semi-supervised settings.
- We verified CGSGM-SC's effectiveness in improving CGSGM, especially in semi-supervised settings.

## 2 BACKGROUND

Consider a data distribution $p(\boldsymbol{x})$ where $\boldsymbol{x} \in \mathbb{R}^d$. The purpose of an SGM is to generate samples from $p(\boldsymbol{x})$ via the information contained in the score function $\nabla_{\boldsymbol{x}} \log p(\boldsymbol{x})$, which is learned from

data. We first introduce how a diffusion process can be combined with learning a score function to effectively sample from $p(\boldsymbol{x})$ in Section 2.1. Next, a comprehensive review of works that have extended SGMs to conditional SGMs is presented in Section 2.2, including those that incorporates classifier regularization for CGSGMs. Finally, JEM (Grathwohl et al., 2020) is introduced in Section 2.3, highlighting its role in inspiring our proposed methodology.

## 2.1 SCORE-BASED GENERATIVE MODELING BY DIFFUSION

Song et al. (2021) propose to model the transition from a known prior distribution $p_T(\boldsymbol{x})$, typically a multivariate gaussian noise, to an unknown target distribution $p_0(\boldsymbol{x}) = p(\boldsymbol{x})$ using the markov chain described by the following stochastic differential equation (SDE):

$$d\boldsymbol{x} = \big[f(\boldsymbol{x}, t) - g(t)^2 s(\boldsymbol{x}, t)\big]dt + g(t)d\bar{\boldsymbol{w}}, \tag{1}$$

where $\bar{\boldsymbol{w}}$ is a standard Wiener process when the timestep flows from $T$ back to 0, $s(\boldsymbol{x}, t) = \nabla_{\boldsymbol{x}} \log p_t(\boldsymbol{x})$ denotes a time-dependent score function, and $f(\boldsymbol{x}, t)$ and $g(t)$ are some prespecified functions that describe the overall movement of the distribution $p_t(\boldsymbol{x})$. The score function is learned by optimizing the following time-generalized denoise score matching (DSM) (Vincent, 2011) loss

$$\mathcal{L}_{DSM}(\boldsymbol{\theta}) = \mathbb{E}_t \left[ \lambda(t) \mathbb{E}_{\boldsymbol{x}_t, \boldsymbol{x}_0} \left[ \frac{1}{2} \|s(\boldsymbol{x}_t, t; \boldsymbol{\theta}) - s_t(\boldsymbol{x}_t|\boldsymbol{x}_0)\|_2^2 \right] \right], \tag{2}$$

where $t$ is selected uniformly between 0 and $T$, $\boldsymbol{x}_t \sim p_t(\boldsymbol{x})$, $\boldsymbol{x}_0 \sim p_0(\boldsymbol{x})$, $s_t(\boldsymbol{x}_t|\boldsymbol{x}_0)$ denotes the score function of the noise distribution $p_t(\boldsymbol{x}_t|\boldsymbol{x}_0)$, which can be calculated using the prespecified $f(\boldsymbol{x}, t)$ and $g(t)$, and $\lambda(t)$ is a weighting function that balances the loss of different timesteps. In this paper, we use the drift $f(\boldsymbol{x}, t)$, dispersion $g(t)$, and weighting $\lambda(t)$ functions from the original VE-SDE framework (Song et al., 2021). A more detailed introduction on learning the score function and sampling through SDEs is described in Appendix B.

## 2.2 CONDITIONAL SCORE-BASED GENERATIVE MODELS

In conditional SGMs, we are given labeled data $\{(\boldsymbol{x}_m, y_m)\}_{m=1}^M$ in addition to unlabeled data $\{\boldsymbol{x}_n\}_{n=M+1}^{M+N}$, where $y_m \in \{1, 2, \dots, K\}$ denotes the class label. The case of $N = 0$ is called the fully-supervised setting; in this paper, we consider the semi-supervised setting with $N > 0$, with a particular focus on the challenging scenario where $\frac{M}{N+M}$ is small. The goal of conditional SGMs is to learn the conditional score function $\nabla_{\boldsymbol{x}} \log p(\boldsymbol{x}|y)$ and then generate samples from $p(\boldsymbol{x}|y)$, typically using a diffusion process as discussed in Section 2.1 and Appendix B.2.

One approach for conditional SGMs is classifier-free SGM (Dhariwal & Nichol, 2021; Ho & Salimans, 2021), which parameterizes its model with a joint architecture such that the class labels $y$ can be included as inputs. Classifier-free guidance (Ho & Salimans, 2021), also known as CFG, additionally uses a null token $y_{\text{NIL}}$ to indicate unconditional score calculation, which is linearly combined with conditional score calculation for some specific $y$ to form the final estimate of $s(\boldsymbol{x}|y)$. CFG is a state-of-the-art conditional SGM in the fully-supervised setting. Nevertheless, as we shall show in our experiments, its performance drops significantly in the semi-supervised setting, as the conditional parts of CFG may lack sufficient labeled data during training.

Another popular family of conditional SGM is CGSGM. Under this framework, we decompose the conditional score function using Bayes' theorem (Song et al., 2021; Dhariwal & Nichol, 2021):

$$\nabla_{\boldsymbol{x}} \log p(\boldsymbol{x}|y) = \nabla_{\boldsymbol{x}}[\log p(\boldsymbol{x}) + \log p(y|\boldsymbol{x}) - \log p(y)] = \nabla_{\boldsymbol{x}} \log p(\boldsymbol{x}) + \nabla_{\boldsymbol{x}} \log p(y|\boldsymbol{x}) \tag{3}$$

The $\log p(y)$ term can be dropped because it is not a function of $\boldsymbol{x}$ and is thus of gradient 0. The decomposition shows that conditional generation can be achieved by an unconditional SGM that learns the score function $\nabla_{\boldsymbol{x}} \log p(\boldsymbol{x})$ plus an extra conditional gradient term $\nabla_{\boldsymbol{x}} \log p(y|\boldsymbol{x})$.

The vanilla classifier-guidance (CG) estimates $\nabla_{\boldsymbol{x}} \log p(y|\boldsymbol{x})$ with an auxiliary classifier trained from the cross-entropy loss on the labeled data and learns the unconditional score function by the denoising score matching loss $\mathcal{L}_{\text{DSM}}$, which in principle can be applied on unlabeled data along with labeled data. Nevertheless, the classifier within the vanilla CG approach is known to be potentially overconfident (Lee et al., 2018; Müller et al., 2019; Mukhoti et al., 2020; Grathwohl et al., 2020) in its predictions, which in turn results in inaccurate gradients. This can mislead the conditional generation process and decrease class-conditional generation quality.

Dhariwal & Nichol (2021) propose to address the issue by post-processing the term $\nabla_{\boldsymbol{x}} \log p(y|\boldsymbol{x})$ with a scaling parameter $\lambda_{CG} \neq 1$.

$$\nabla_{\boldsymbol{x}} \log p(\boldsymbol{x}|y) = \nabla_{\boldsymbol{x}} \log p(\boldsymbol{x}) + \lambda_{CG} \nabla_{\boldsymbol{x}} \log p(y|\boldsymbol{x}; \boldsymbol{\phi}), \tag{4}$$

where $p(y|\boldsymbol{x}; \boldsymbol{\phi})$ is the posterior probability distribution outputted by a classifier parameterized by $\boldsymbol{\phi}$. Increasing $\lambda_{CG}$ sharpens the distribution $p(y|\boldsymbol{x}; \boldsymbol{\phi})$, guiding the generation process to produce less diverse but higher fidelity samples. While the tuning heuristic is effective in improving the vanilla CG approach, it is not backed by sound theoretical explanations.

Other attempts to regularize the classifier *during training* for resolving the issue form a promising research direction. For instance, CGSGM with denoising likelihood score matching (CG-DLSM; Chao et al., 2022) presents a regularization technique that employs the DLSM loss below formulated from the classifier gradient $\nabla_{\boldsymbol{x}} \log p(y, t|\boldsymbol{x}; \boldsymbol{\phi})$ and unconditional score function $s_t(\boldsymbol{x})$.

$$\mathcal{L}_{\text{DLSM}}(\boldsymbol{\phi}) = \mathbb{E}_t \left[ \lambda(t) \mathbb{E}_{\boldsymbol{x}_t, \boldsymbol{x}_0} \left[ \frac{1}{2} \| \nabla_{\boldsymbol{x}} \log p(y, t|\boldsymbol{x}_t; \boldsymbol{\phi}) + s_t(\boldsymbol{x}_t) - s_t(\boldsymbol{x}_t|\boldsymbol{x}_0) \|_2^2 \right] \right], \tag{5}$$

where the unconditional score function $s_t(\boldsymbol{x})$ is estimated via an unconditional SGM $s(\boldsymbol{x}_t, t; \boldsymbol{\theta})$. The CG-DLSM authors (Chao et al., 2022) prove that Eq. 5 can calibrate the classifier to produce more accurate gradients $\nabla_x \log p(y|\boldsymbol{x})$.

Robust CGSGM (Kawar et al., 2022), in contrast to CG-DLSM, does not regularize by modeling the unconditional distribution. Instead, robust CGSGM leverages existing techniques to improve the robustness of the classifier against *adversarial* perturbations. Robust CGSGM applies a gradient-based adversarial attack to the $\boldsymbol{x}_t$ generated during the diffusion process, and uses the resulting adversarial example to make the classifier more robust.

## 2.3 REINTERPRETING CLASSIFIERS AS ENERGY-BASED MODELS (EBMS)

Our proposed methodology draws inspiration from JEM (Grathwohl et al., 2020), which shows that reinterpreting classifiers as EBMs and enforcing regularization with related objectives helps classifiers to capture more accurate probability distributions. EBMs are models that estimate energy functions $E(\boldsymbol{x})$ of distributions (LeCun et al., 2006), which satisfies $\log p(\boldsymbol{x}) = -E(\boldsymbol{x}) + \log \int_{\boldsymbol{x}} \exp(E(\boldsymbol{x})) d\boldsymbol{x}$. Given the logits of a classifier to be $f(\boldsymbol{x}, y; \boldsymbol{\phi})$, the estimated joint distribution can be written as $p(\boldsymbol{x}, y; \boldsymbol{\phi}) = \frac{\exp(f(\boldsymbol{x}, y; \boldsymbol{\phi}))}{Z(\boldsymbol{\phi})}$, where $\exp(\cdot)$ means exponential and $Z(\boldsymbol{\phi}) = \int_{\boldsymbol{x}, y} \exp(f(\boldsymbol{x}, y; \boldsymbol{\phi})) d\boldsymbol{x} \, dy$. After that, the energy function $E(\boldsymbol{x}; \boldsymbol{\phi})$ can be obtained by

$$E(\boldsymbol{x}; \boldsymbol{\phi}) = -\log \Sigma_y \exp(f(\boldsymbol{x}, y; \boldsymbol{\phi})) \tag{6}$$

Then, losses used to train EBMs can be seamlessly leveraged in JEM to regularize the classifier, such as the typical EBM loss $\mathcal{L}_{\text{EBM}} = \mathbb{E}_{p(\boldsymbol{x})} [-\log p(\boldsymbol{x}; \boldsymbol{\phi})]$. JEM uses MCMC sampling for computing the loss, and is shown to result in a well-calibrated classifier in their empirical study. The original JEM work (Grathwohl et al., 2020) also reveals that classifiers can potentially be used as a reasonable generative model, but its (unconditional) generation performance is knowingly worse than state-of-the-art SGMs (Song et al., 2021).

## 3 THE PROPOSED SELF-CALIBRATION METHODOLOGY

In this work, we consider CGSGMs under the diffusion generation process as discussed in Section 2.1. Such CGSGMs require learning an unconditional SGM, which is assumed to be trained with denoising score matching (DSM; Vincent, 2011) due to its close relationship with the diffusion process. Such CGSGMs also require a time-dependent classifier that models $p_t(y|\boldsymbol{x})$ instead of $p(y|\boldsymbol{x})$, which can be done by applying a time-generalized cross-entropy loss.

Section 2.2 has illustrated the importance of regularizing the (time-dependent) classifier to prevent it from misguiding the conditional generation process. One naive thought is to use established regularization mechanisms, such as label-smoothing and Jacobian regularization (Hoffman et al., 2019). Those regularization possibilities that are less attached to the underlying distribution will be studied in Section 4. Our proposed regularization loss, inspired by the success of CG-DLSM Chao et al. (2022), attempts to connect with the underlying distribution better.

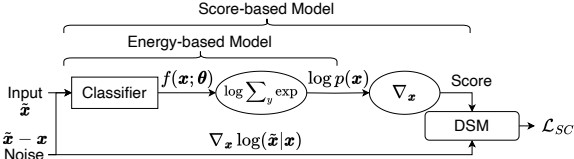

Figure 2: Calculation of proposed self-calibration loss

## 3.1 FORMULATION OF SELF-CALIBRATION LOSS

We extend JEM (Grathwohl et al., 2020) to connect the time-dependent classifier to the underlying distribution. In particular, we reinterpret the classifier as a time-dependent EBM. The interpretation allows us to obtain a time-dependent version of $p_t(\boldsymbol{x})$ within the classifier, which can be used to obtain a classifier-*internal* version of the score function. Then, instead of regularizing the classifier by the EBM loss $-\log p_t(\boldsymbol{x})$ like JEM (as unsuccessfully studied in Appendix E), we propose to regularize by score function $\nabla_{\boldsymbol{x}} \log p_t(\boldsymbol{x})$ instead.

Under the EBM interpretation, the energy function is $E(\boldsymbol{x}, t; \boldsymbol{\phi}) = -\log \Sigma_y \exp(f(\boldsymbol{x}, y, t; \boldsymbol{\phi}))$, where $f(\boldsymbol{x}, y, t; \boldsymbol{\phi})$ is the output logits of the time-dependent classifier. Then, the internal time-dependent unconditional score function is $s^c(\boldsymbol{x}, t; \boldsymbol{\phi}) = \nabla_{\boldsymbol{x}} \log \Sigma_y \exp(f(\boldsymbol{x}, y, t; \boldsymbol{\phi}))$, where $s^c$ is used instead of $s$ to indicate that the unconditional score is computed *within* the classifier. Then, we adopt the standard DSM technique in Eq. 2 to "train" the internal score function, forcing it to follow its physical meaning during the diffusion process. The resulting self-calibration loss can then be defined as

$$\mathcal{L}_{\mathrm{SC}}(\boldsymbol{\phi}) = \mathbb{E}_t \left[ \lambda(t) \mathbb{E}_{\boldsymbol{x}_t, \boldsymbol{x}_0} \left[ \frac{1}{2} \left\| s^c(\boldsymbol{x}_t, t; \boldsymbol{\phi}) - s_t(\boldsymbol{x}_t | \boldsymbol{x}_0) \right\|_2^2 \right] \right], \tag{7}$$

where $\boldsymbol{x}_t \sim p_t$, $\boldsymbol{x}_0 \sim p_0$, and $s_t(\boldsymbol{x}_t | \boldsymbol{x}_0)$ denotes the score function of the noise centered at $\boldsymbol{x}_0$.

Fig. 2 summarizes the calculation of the proposed SC loss. Note that in practice, $t$ is uniformly sampled over $[0, T]$. After the self-calibration loss is obtained, it is mixed with the cross-entropy loss $\mathcal{L}_{\mathrm{CE}}$ to train the classifier. The total loss can be written as:

$$\mathcal{L}_{\mathrm{CLS}}(\boldsymbol{\phi}) = \mathcal{L}_{\mathrm{CE}}(\boldsymbol{\phi}) + \lambda_{SC} \mathcal{L}_{\mathrm{SC}}(\boldsymbol{\phi}), \tag{8}$$

where $\lambda_{SC}$ is a tunable hyper-parameter. The purpose of self-calibration is to cause the classifier to more accurately estimate the score function of the underlying data distribution, implying that the underlying data distribution itself is also more accurately estimated. As a result, the gradients of the classifiers are more aligned with the ground truth. After self-calibration, the classifier is then used in CGSGM to guide an unconditional SGM for conditional generation. Note that since our approach regularizes the classifier during training while classifier gradient scaling (Eq. 4) is done during sampling, we can easily combine the two techniques to enhance performance.

## 3.2 COMPARISON WITH RELATED REGULARIZATION METHODS

This section provides a comparative analysis of the regularization methods employed in DLSM (Chao et al., 2022), robust classifier guidance (Kawar et al., 2022), JEM Grathwohl et al. (2020), and our proposed self-calibration loss.

**DLSM (Chao et al., 2022)** DLSM and our proposed method both regularize the classifier to align better with unconditional SGMs' view of the underlying distribution. DLSM achieves this by relying on the help of an *external* trained SGM, whereas self-calibration regularizes the classifier by using a classifier-*internal* SGM. Furthermore, DLSM loss only utilizes labeled data during training, while our method is able to make use of unlabeled data in addition to labeled data.

**Robust CGSGM (Kawar et al., 2022)** Robust CGSGM proposes to regularize classifiers with gradient-based adversarial training without explicitly aligning with the distribution, in contrast to

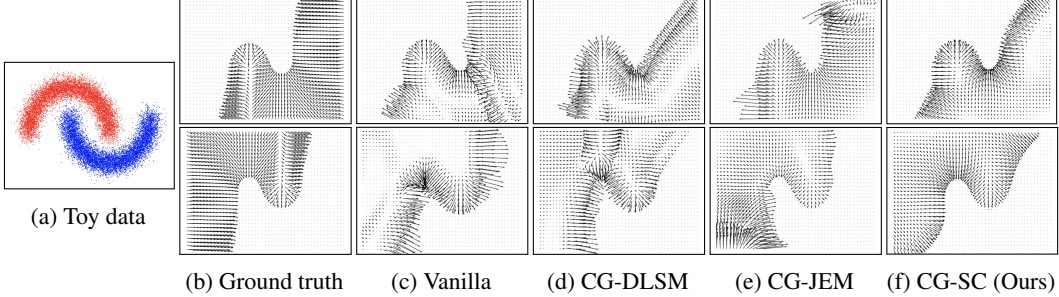

Figure 3: Gradients of classifiers $\nabla_{\boldsymbol{x}} \log p(y|\boldsymbol{x})$ for toy dataset. The upper row contains the gradients for class 1 (red) and the lower contains the gradients for class 2 (blue). (a) Real data distribution. (b) Ground truth classifier gradients. Gradients estimated by (c) Vanilla CG, (d) CG-DLSM, (e) CG-JEM, and (f) CG with proposed self-calibration.

our self-calibration method, where direct calibration of the classifier-estimated score function is employed. Although adversarial robustness is correlated with more interpretable gradients (Tsipras et al., 2019), EBMs (Zhu et al., 2021), and generative modeling (Santurkar et al., 2019), the theoretical foundation for whether adversarially robust classifiers accurately estimate underlying distributions remains ambiguous.

**JEM (Grathwohl et al., 2020)**  JEM and the proposed self-calibration both interpret classifiers as unconditional EBMs, and self-calibration further extends the interpretation to unconditional SGMs. The training stage of EBM that incorporates MCMC sampling is known to be unstable and time-consuming, whereas self-calibration precludes the need for sampling during training and substantially improves both stability and efficiency. Even though one can mitigate the instability issue by increased sampling steps and additional hyperparameter tuning, doing so largely lengthens training times.

**CG-JEM**  In contrast to the previous paragraph, this paragraph discusses the incorporation of JEM into the time-dependent CGSGM framework. Coupling EBM training with additional training objectives is known to introduce increased instability, especially for time-dependent classifiers considering it is more difficult to generate meaningful time-dependent data through MCMC sampling. For example, in our naive implementation of time-dependent JEM, it either (1) incurs high instability (loss diverges within $10,000$ steps in all 10 runs; requires $4.23$s per step) or (2) poses unaffordable resource consumption requirements (requires $20$s per step, approximately $50$ days in total). In comparison, our proposed method only requires $0.75$s per step. Details of our attempts to incorporate JEM into the current framework are provided in Appendix E.

### 3.3 2D TOY DATASET

Following DLSM (Chao et al., 2022), we use a 2D toy dataset containing two classes to demonstrate the effects of self-calibration loss and visualize the training results. The data distribution is shown in Fig. 3a, where the two classes are shown in two different colors. After training the classifiers on the toy dataset with different methods, we plot the gradients $\nabla_{\boldsymbol{x}} \log p(y|\boldsymbol{x})$ at minimum timestep $t = 0$ estimated by the classifiers and compare them with the ground truth. Additional quantitative measurements of the toy dataset are included in Appendix F.

Figure 3 shows the ground truth classifier gradient (Fig. 3b) and the gradients estimated by classifiers trained using different methods. Unregularized classifiers produce gradients that contain rapid changes in magnitude across the 2D space, with frequent fluctuations and mismatches with the ground truth. Such fluctuations can impede the convergence of the reverse diffusion process to a stable data point, leading SGMs to generate noisier samples. Moreover, the divergence from the ground truth gradient can misguide the SGM, leading to generation of samples from incorrect classes. Unregularized classifiers also tend to generate large gradients near the distribution borders and tiny gradients elsewhere. This implies that when the sampling process is heading toward the incorrect class, such classifiers are not able to "guide" the sampling process back towards the desired class.

In comparison, the introduction of various regularization techniques such as DLSM, JEM, and the proposed self-calibration results in estimated gradients that are more stable, continuous across the 2D space, and better aligned with the ground truth. This stability brings about a smoother generation process and the production of higher-quality samples. In Section 4, we will further examine the generation performance of various methods with the CIFAR-10 and CIFAR-100 dataset.

### 3.4 USING SELF-CALIBRATION LOSS ON SEMI-SUPERVISED LEARNING

In this work, we also explore the benefit of self-calibration loss in the semi-supervised setting, where only a small proportion of data are labeled. In the original classifier guidance, the classifiers are trained only on labeled data. The lack of labels in the semi-supervised setting constitutes a greater challenge to learning an unbiased classifier. With self-calibration, we better utilize the large amount of unlabeled data by calculating the self-calibration loss with all data.

To incorporate the loss and utilize unlabeled data during training, we change the way $\mathcal{L}_{\mathrm{CLS}}$ is calculated from Eq. 8. As illustrated in Fig. 1, the entire batch of data is used to calculate $\mathcal{L}_{\mathrm{SC}}$, whereas only the labeled data is used to calculate $\mathcal{L}_{\mathrm{CE}}$. During training, we observe that when the majority is unlabeled data, the cross-entropy loss does not converge to a low-and-steady stage if the algorithm randomly samples from all training data. As this may be due to the low percentage of labeled data in each batch, we change the way we sample batches by always ensuring that exactly half of the data is labeled. Appendix C summarizes the semi-supervised training process of the classifier.

Note that even though the classifier is learning a time-generalized classification task, we can still make it perform as an ordinary classifier that classifies the unperturbed data by setting the input timestep $t = 0$. This greatly facilitates the incorporation of common semi-supervised classification methods such as pseudo-labeling (Lee, 2013), self-training, and noisy student (Xie et al., 2020). Integrating semi-supervised classification methodologies is an interesting future research direction, and we reserve the detailed exploration of this topic for future studies.

## 4 EXPERIMENTS

We tested our method on a toy dataset (Section 3.3) to provide a high-level view of how SC loss improves classifiers in terms of producing accurate gradients. In the following sections, we test our methods on the CIFAR-10 and CIFAR-100 datasets for image generation. We demonstrate that our method improves generation both conditionally and unconditionally with different percentages of labeled data (Section 4.2). Randomly selected images of CGSGM before and after self-calibration on CIFAR-10 are shown in Appendix I. Addtional experimental details are included in Appendix D.

### 4.1 EXPERIMENTAL SETUP

**Evaluation metrics** We evaluated the class-conditional performance of our methods using intra-FID, which measures the average FID for each class, and generation accuracy, which uses a pre-trained ViT (Dosovitskiy et al., 2021) classifier to determine whether the samples are generated in the correct class, in addition to commonly used metrics Frechet inception distance (FID; Heusel et al., 2017) and inception score (IS; Salimans et al., 2016). The test accuracy of the pre-trained ViT is 98.52% on CIFAR-10. Note that nearly-accurate classification is desired for meaningful evaluation of the generation accuracy, but we were unable to locate such a classifier for the CIFAR-100 dataset. Therefore, generation accuracy is not included for the CIFAR-100 dataset.

**Baseline methods** **CG**: vanilla classifier guidance; **CG-DLSM**: classifier guidance with DLSM loss (Chao et al., 2022); **CG-LS**: classifier guidance with label smoothing; **CG-JR**: classifier guidance with Jacobian regularization (Hoffman et al., 2019); **Cond**: conditional SGMs by conditional normalization (Dhariwal & Nichol, 2021); **CFG-labeled**: CFG (Ho & Salimans, 2021) using only labeled data; **CFG-all**: CFG using all data to train the unconditional part of the model.

### 4.2 RESULTS

Table 1 and Fig. 4 present the performance of all methods when applied to various percentages of labeled data. Notice that this includes the fully-supervised setting when 100% of data are la-

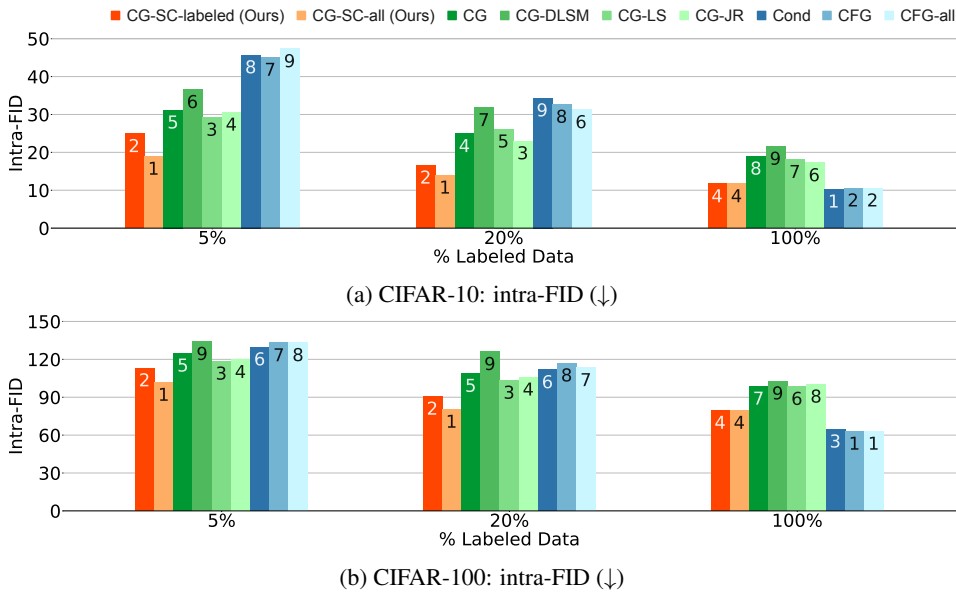

Figure 4: Results of class-conditional generation. The performance ranking is shown for reference.

beled. **CG-SC-labeled** implies that self-calibration is applied only on labeled data whereas **CG-SC-all** implies that self-calibration is applied on all data. In Table 1, **bold** entries represent the best performance among all methods, and underlined entries represent the best performance among classifier-guidance-based methods. Supplementary experimental results on CIFAR-10 are included in Appendix A.

**Classifier-Free SGMs (CFSGMs)** The first observation from the results is that CFSGMs, including Cond, CFG-labeled, and CFG-all, consistently excel in generation accuracy (Table 1a). However, when the quantity of labeled data falls below 20%, we witness a significant performance drop in these models (Table 1 and Fig. 4). CFSGMs, while generating high-quality images, tend to lack diversity when working with fewer labeled data. This occurs mainly due to the lack of sufficient labeled data in the training phase, causing them to generate samples that closely mirror the distribution of only the labeled data, as opposed to that of all data. This shows that such methods only learn to generate samples highly correlated to the labeled data, failing to capture the diversity of the entire data; they thus exhibit poor performance when evaluated with related measures (FID, intra-FID).

**Classifier-Guided SGMs (CGSGMs)** CGSGMs demonstrate superior performance in semi-supervised settings, as they leverage both labeled and unlabeled data during training. CGSGMs exhibit consistent performance in terms of FID and IS across various percentages of labeled data (Table 1). Notably, when unlabeled data is in the majority, we observe a 16% drop in generation accuracy on the CIFAR-10 dataset (Table 1a). Despite this, the intra-FID of CG significantly outperforms that of CFSGMs on both datasets. This superior performance is consistent across all CG-based methods, including the vanilla CGSGM and CGSGM with various regularization techniques. This shows that CGSGM-based methods are preferable under semi-supervised settings, as they capture the diversity of training data through higher utilization of unlabeled data.

**Regularized CGSGMs vs Vanilla CGSGM** Basic regularization methods like label-smoothing and Jacobian regularization (Hoffman et al., 2019) show marginal improvement over vanilla CGSGM. This points out that although these methods mitigate overfitting on training data, the constraints they enforce do not align with SGMs, limiting the benefit of including such methods. CG-DLSM (Chao et al., 2022), on the other hand, always achieves great unconditional generation performance. However, its class-conditional performance suffers from a significant performance drop as labeled data is lowered from 100% to 5% (Table 1). Incorporating the proposed self-calibration with labeled data substantially improves conditional metrics (Table 1). Notably, CG-SC (Ours) consistently achieves the best intra-FID and generation accuracy among all CG-based methods. On

Table 1: Sample quality comparison of all methods with various percentages of labeled data. **Bold**: best performance among all methods; underlined: best performance among CG-based methods.

(a) Results of semi-supervised settings on CIFAR-10 dataset

| Method | 5% labeled data | | | | 20% labeled data | | | | 100% labeled data | | | |
|---|---|---|---|---|---|---|---|---|---|---|---|---|
| | intra-FID ($\downarrow$) | Acc ($\uparrow$) | FID ($\downarrow$) | IS ($\uparrow$) | intra-FID ($\downarrow$) | Acc ($\uparrow$) | FID ($\downarrow$) | IS ($\uparrow$) | intra-FID ($\downarrow$) | Acc ($\uparrow$) | FID ($\downarrow$) | IS ($\uparrow$) |
| CG-SC-labeled (Ours) | 24.93 | 0.525 | 2.84 | 9.78 | 16.62 | 0.672 | 2.75 | 9.83 | 11.70 | 0.829 | 2.23 | 9.82 |
| CG-SC-all (Ours) | **18.95** | 0.676 | 2.72 | 9.95 | **13.97** | 0.752 | 2.63 | 9.94 | 11.70 | 0.829 | 2.23 | 9.82 |
| CG | 31.17 | 0.448 | 2.61 | 9.98 | 24.94 | 0.530 | 3.09 | 9.92 | 18.99 | 0.611 | 2.48 | 9.88 |
| CG-DLSM | 36.55 | 0.354 | 2.18 | 9.76 | 31.78 | 0.419 | 2.10 | 9.91 | 21.59 | 0.564 | 2.36 | 9.92 |
| CG-LS | 29.24 | 0.466 | 2.62 | 9.92 | 26.15 | 0.522 | 4.18 | 9.98 | 18.10 | 0.636 | 2.15 | 9.98 |
| CG-JR | 30.59 | 0.455 | 2.80 | 9.84 | 23.03 | 0.552 | 2.49 | **10.04** | 17.24 | 0.643 | 2.17 | 9.89 |
| Cond | 45.73 | 0.959 | 15.57 | 9.87 | 34.36 | 0.927 | 19.77 | 8.82 | 10.29 | 0.970 | **2.13** | **10.06** |
| CFG-labeled | 45.07 | 0.950 | 15.31 | **10.20** | 32.66 | 0.893 | 18.48 | 8.93 | 10.58 | **0.971** | 2.28 | 10.05 |
| CFG-all | 47.33 | **0.964** | 16.57 | 9.89 | 31.24 | **0.936** | 17.37 | 9.15 | 10.58 | **0.971** | 2.28 | 10.05 |

(b) Results of semi-supervised settings on CIFAR-100 dataset

| Method | 5% labeled data | | | 20% labeled data | | | 100% labeled data | | |
|---|---|---|---|---|---|---|---|---|---|
| | intra-FID ($\downarrow$) | FID ($\downarrow$) | IS ($\uparrow$) | intra-FID ($\downarrow$) | FID ($\downarrow$) | IS ($\uparrow$) | intra-FID ($\downarrow$) | FID ($\downarrow$) | IS ($\uparrow$) |
| CG-SC-labeled (Ours) | 113.21 | 4.80 | 12.06 | 90.76 | 3.74 | 12.25 | 79.57 | 3.70 | 11.69 |
| CG-SC-all (Ours) | **101.75** | 4.31 | 12.49 | **80.42** | **3.60** | **12.60** | 79.57 | 3.70 | 11.69 |
| CG | 124.92 | 5.24 | 12.06 | 108.86 | 4.10 | 12.02 | 98.72 | 3.83 | 11.89 |
| CG-DLSM | 134.11 | 4.46 | 11.94 | 126.12 | 7.24 | 11.40 | 102.85 | 3.85 | 11.76 |
| CG-LS | 118.52 | 4.18 | 12.34 | 103.39 | 3.70 | 12.60 | 98.53 | 3.39 | 12.09 |
| CG-JR | 119.78 | 4.64 | 12.32 | 105.91 | 3.92 | 12.48 | 100.34 | 3.50 | 11.97 |
| Cond | 129.82 | 10.58 | **14.90** | 111.73 | 29.45 | 9.98 | 64.77 | 3.02 | 12.97 |
| CFG-labeled | 133.03 | 11.25 | 14.81 | 117.09 | 32.68 | 9.75 | **63.03** | **2.60** | **13.61** |
| CFG-all | 133.18 | 10.68 | 14.87 | 113.38 | 30.84 | 10.09 | **63.03** | **2.60** | **13.61** |

average, CG-SC-all improves intra-FID by 10.16 and 23.59 over CG on CIFAR-10 and CIFAR-100, respectively. Furthermore, self-calibration increases generation accuracy on CIFAR-10 by up to 23% (Table 1a). These results demonstrate that self-calibration enables the classifier to estimate the class-conditional distribution more accurately, even when labeled data is limited.

**Leverage unlabeled data for semi-supervised conditional generation** Intuitively, we expect incorporating unlabeled data into the computation of SC loss to enhance the quality of conditional generation, because the classifier can exploit additional information from unlabeled data during the training phase. As the proportion of unlabeled data increases, we expect this benefit of leveraging unlabeled data to become more significant. As our experimental results indicate in Fig. 4, conditional metrics do not differ greatly in the fully-supervised scenario. However, when the percentage of labeled data falls below 20%, the utilization of unlabeled data significantly improves intra-FID and accuracy (Table 1a). Specifically, with only 5% of the data labeled, intra-FID, and generation accuracy are improved by 12.22 and 22.8% over the original CG on CIFAR-10. These results confirm our expectation that as the percentage of labeled data decreases, the beneficial impact of utilizing unlabeled data increases.

## 5 CONCLUSION

We tackle the overfitting issue for the classifier within CGSGM from a novel perspective: self-calibration. We leverage the EBM interpretation like JEM to reveal that the classifier is internally an unconditional score estimator and design a loss with the DSM technique to calibrate the internal estimation. This self-calibration loss regularizes the classifier directly towards better scores without relying on an external score estimator. We demonstrate three immediate benefits of the proposed self-calibrating CGSGM approach. Using a standard synthetic dataset, we show that the scores computed using this approach are indeed closer to the ground-truth scores. Second, across all percentages of labeled data, the proposed approach outperforms the existing CGSGM. Last, our empirical study justifies that when compared to other conditional SGMs, the proposed approach consistently achieves the best intra-FID in the focused semi-supervised settings by seamlessly leveraging the power of unlabeled data. The benefits establish the rich potential of the proposed approach.

# 6 REPRODUCIBILITY

We included source code for the implemented self-calibration loss in the supplementary materials. A readme file is included to provide instructions on the usage of the code. One can also refer to Appendix C for algorithmic details and Appendix D for experimental details to implement the proposed self-calibration loss.

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

## A  SUPPLEMENTARY EXPERIMENTAL RESULTS ON CIFAR-10

### A.1  ADDITIONAL SEMI-SUPERVISED LEARNING SETTINGS

In Section 4, we discussed the generative performance using 5%, 20%, and 100% labeled data from CIFAR-10. In this section, we provide further results for scenarios where 40%, 60%, and 80% of the data is labeled.

Table 2: Sample quality comparison of all methods with 40%, 60%, and 80% labeled data. **Bold**: best performance among all methods; underlined: best performance among CG-based methods.

| Method | 40% labeled data | | | | 60% labeled data | | | | 80% labeled data | | | |
|---|---|---|---|---|---|---|---|---|---|---|---|---|
| | intra-FID (↓) | Acc (↑) | FID (↓) | IS (↑) | intra-FID (↓) | Acc (↑) | FID (↓) | IS (↑) | intra-FID (↓) | Acc (↑) | FID (↓) | IS (↑) |
| CG-SC-labeled (Ours) | **12.08** | 0.862 | 2.78 | 10.00 | 11.65 | 0.850 | 2.37 | 9.91 | 11.86 | 0.823 | 2.24 | 9.78 |
| CG-SC-all (Ours) | 12.67 | 0.809 | 2.72 | **10.04** | 12.22 | 0.810 | 2.42 | 9.95 | 12.47 | 0.788 | 2.25 | 9.83 |
| CG | 18.31 | 0.628 | 2.42 | 9.95 | 16.94 | 0.656 | 2.35 | 10.03 | 20.15 | 0.609 | 3.30 | 9.76 |
| CG-DLSM | 29.33 | 0.457 | 2.35 | 9.85 | 23.52 | 0.531 | **2.15** | 9.83 | 21.76 | 0.563 | 2.30 | 9.96 |
| CG-LS | 17.89 | 0.638 | **2.32** | 9.95 | 17.72 | 0.638 | 2.27 | 9.91 | 22.30 | 0.576 | 2.40 | 9.84 |
| CG-JR | 18.63 | 0.625 | 2.43 | 10.01 | 19.05 | 0.609 | 2.25 | 10.06 | 18.36 | 0.622 | **2.15** | 9.90 |
| Cond | 13.65 | 0.962 | 4.36 | 9.94 | **10.93** | 0.968 | 2.55 | 10.00 | **10.61** | 0.968 | 2.37 | 10.03 |
| CFG-labeled | 13.93 | 0.948 | 4.59 | 9.84 | 11.28 | 0.966 | 2.73 | **10.12** | 10.75 | **0.972** | 2.48 | **10.09** |
| CFG-all | 13.43 | **0.970** | 4.30 | 9.98 | 11.38 | **0.972** | 2.83 | 10.05 | 10.94 | 0.970 | 2.50 | 10.03 |

Table 2 presents the results, further confirming the observations made in Section 4.2. The CFS-GMs consistently exhibit high generation accuracy, but suffer from significant performance drop as the labeled data percentage decreases. Conversely, the CGSGMs maintain stable performance across various settings. Furthermore, our proposed CG-SC consistently outperforms other CG-based methodologies in terms of intra-FID and generation accuracy.

### A.2  EVALUATION OF EXPECTED CALIBRATION ERROR

Beyond the generative performance metrics, we present the Expected Calibration Error (ECE) to assess the calibration of classifiers regarding accurate probability estimation. ECE serves as a metric that evaluates the alignment of a classifier's confidence with its prediction accuracy. The classifier's confidence is defined as:

$$\text{conf}(\boldsymbol{x}) = \max_y p(y|\boldsymbol{x}) = \max_y \frac{\exp(f(\boldsymbol{x}, y; \boldsymbol{\phi}))}{\sum_{y'} \exp(f(\boldsymbol{x}, y'; \boldsymbol{\phi}))},$$

where $f(\boldsymbol{x}, y; \boldsymbol{\phi})$ is the classifier's logits. We then divide the classifier's predictions based on confidence into several buckets. The average absolute difference between the confidence and prediction accuracy is calculated for each bucket. Then, given a labeled test set $D_t = \{(\boldsymbol{x}_m, y_m)\}_{m=1}^M$, ECE is defined as:

$$\text{ECE} = \sum_{i=1}^N \frac{|B_i|}{|D_t|} \cdot \left| \text{Acc}(B_i) - \frac{1}{|B_i|} \sum_{\boldsymbol{x} \in B_i} \text{conf}(\boldsymbol{x}) \right|,$$

where $N$ is the number of buckets, $B_i = \left\{ \boldsymbol{x} | \text{conf}(\boldsymbol{x}) \in [\frac{i-1}{N}, \frac{i}{N}) \right\}$, $\text{Acc}(B_i)$ is the averaged classification accuracy of $B_i$, and $\frac{1}{|B_i|} \sum_{\boldsymbol{x} \in B_i} \text{conf}(\boldsymbol{x})$ is the averaged confidence of $B_i$.

Table 3: Expected calibration error (↓) of all methods with various percentages of labeled data

| Method | 5% | 20% | 40% | 60% | 80% | 100% |
|---|---|---|---|---|---|---|
| CG-SC-labeled (Ours) | 0.369 | 0.316 | **0.087** | **0.057** | **0.063** | **0.031** |
| CG-SC-all (Ours) | 0.210 | **0.243** | 0.102 | 0.109 | 0.111 | **0.031** |
| CG | 0.460 | 0.330 | 0.269 | 0.190 | 0.163 | 0.112 |
| CG-DLSM | 0.468 | 0.343 | 0.307 | 0.237 | 0.180 | 0.183 |
| CG-LS | **0.194** | 0.257 | 0.101 | 0.063 | 0.081 | 0.050 |
| CG-JR | 0.407 | 0.348 | 0.279 | 0.225 | 0.183 | 0.173 |

We follow the setup in Grathwohl et al. (2020), setting $N = 20$ for our calculations. The results are shown in Table 3, illustrating the ECE values for all CG-based methods across various percentages of labeled data. Our observations underscore that the self-calibration method consistently enhances classifier ECE in comparison to the vanilla CG and delivers the most superior ECE in most cases. This validates our claim that self-calibrated classifiers offer a more accurate probability estimation.

## A.3 DETAILED CLASS-CONDITIONAL METRICS

The conditional metrics in Section 4.2 are averaged among all classes. Here, we also included the detailed conditional metrics for each class in CIFAR-10.

Table 4: CG-SC-labeled (Ours)

| Class | 5% labeled data intra-FID | Acc | 20% labeled data intra-FID | Acc | 40% labeled data intra-FID | Acc | 60% labeled data intra-FID | Acc | 80% labeled data intra-FID | Acc | 100% labeled data intra-FID | Acc |
|---|---|---|---|---|---|---|---|---|---|---|---|---|
| Airplane | 23.24 | 0.640 | 16.43 | 0.759 | 14.09 | 0.890 | 12.78 | 0.883 | 11.91 | 0.858 | 11.98 | 0.858 |
| Automobile | 18.01 | 0.634 | 10.56 | 0.806 | 9.04 | 0.937 | 8.71 | 0.941 | 8.43 | 0.912 | 8.29 | 0.921 |
| Bird | 31.57 | 0.367 | 21.54 | 0.514 | 13.20 | 0.790 | 13.30 | 0.779 | 14.17 | 0.751 | 14.30 | 0.755 |
| Cat | 27.67 | 0.342 | 19.44 | 0.475 | 14.54 | 0.696 | 14.31 | 0.705 | 15.43 | 0.664 | 14.91 | 0.670 |
| Deer | 28.13 | 0.381 | 18.69 | 0.529 | 11.32 | 0.828 | 10.99 | 0.800 | 11.00 | 0.766 | 10.74 | 0.779 |
| Dog | 27.32 | 0.515 | 20.96 | 0.640 | 14.85 | 0.835 | 15.43 | 0.817 | 16.82 | 0.777 | 16.32 | 0.781 |
| Frog | 30.05 | 0.507 | 18.39 | 0.679 | 12.50 | 0.909 | 12.24 | 0.892 | 12.01 | 0.889 | 12.00 | 0.899 |
| Horse | 21.38 | 0.638 | 16.38 | 0.761 | 13.63 | 0.910 | 11.88 | 0.890 | 10.98 | 0.865 | 11.09 | 0.870 |
| Ship | 19.61 | 0.640 | 12.38 | 0.776 | 9.86 | 0.906 | 9.18 | 0.900 | 9.38 | 0.875 | 8.99 | 0.890 |
| Truck | 22.27 | 0.584 | 11.38 | 0.781 | 7.74 | 0.919 | 7.65 | 0.893 | 8.46 | 0.869 | 8.36 | 0.871 |

Table 5: CG-SC-all (Ours)

| Class | 5% labeled data intra-FID | Acc | 20% labeled data intra-FID | Acc | 40% labeled data intra-FID | Acc | 60% labeled data intra-FID | Acc | 80% labeled data intra-FID | Acc | 100% labeled data intra-FID | Acc |
|---|---|---|---|---|---|---|---|---|---|---|---|---|
| Airplane | 19.95 | 0.805 | 14.88 | 0.823 | 14.66 | 0.858 | 12.72 | 0.852 | 12.75 | 0.809 | 11.98 | 0.858 |
| Automobile | 13.15 | 0.758 | 9.37 | 0.846 | 9.20 | 0.907 | 8.79 | 0.916 | 8.47 | 0.881 | 8.29 | 0.921 |
| Bird | 25.12 | 0.479 | 16.68 | 0.647 | 14.45 | 0.716 | 14.99 | 0.715 | 15.76 | 0.697 | 14.30 | 0.755 |
| Cat | 22.95 | 0.448 | 16.93 | 0.566 | 15.78 | 0.590 | 15.31 | 0.617 | 15.49 | 0.653 | 14.91 | 0.670 |
| Deer | 20.10 | 0.532 | 13.50 | 0.675 | 11.61 | 0.763 | 11.47 | 0.761 | 11.36 | 0.729 | 10.74 | 0.779 |
| Dog | 23.03 | 0.648 | 19.39 | 0.694 | 16.08 | 0.780 | 16.19 | 0.763 | 17.67 | 0.755 | 16.32 | 0.781 |
| Frog | 19.12 | 0.721 | 13.80 | 0.787 | 13.06 | 0.839 | 12.64 | 0.857 | 13.05 | 0.838 | 12.00 | 0.899 |
| Horse | 19.05 | 0.815 | 14.56 | 0.821 | 13.51 | 0.887 | 12.25 | 0.869 | 11.41 | 0.822 | 11.09 | 0.870 |
| Ship | 13.34 | 0.798 | 10.85 | 0.836 | 9.97 | 0.887 | 9.62 | 0.869 | 9.75 | 0.855 | 8.99 | 0.890 |
| Truck | 13.68 | 0.752 | 9.78 | 0.826 | 8.32 | 0.859 | 8.21 | 0.878 | 9.00 | 0.844 | 8.36 | 0.871 |

Table 6: CG

| Class | 5% labeled data intra-FID | Acc | 20% labeled data intra-FID | Acc | 40% labeled data intra-FID | Acc | 60% labeled data intra-FID | Acc | 80% labeled data intra-FID | Acc | 100% labeled data intra-FID | Acc |
|---|---|---|---|---|---|---|---|---|---|---|---|---|
| Airplane | 25.99 | 0.586 | 20.00 | 0.645 | 17.13 | 0.706 | 15.85 | 0.725 | 16.79 | 0.677 | 17.92 | 0.649 |
| Automobile | 27.78 | 0.522 | 19.55 | 0.617 | 13.75 | 0.718 | 12.24 | 0.770 | 13.08 | 0.729 | 12.84 | 0.713 |
| Bird | 39.10 | 0.268 | 34.35 | 0.345 | 23.40 | 0.485 | 22.01 | 0.521 | 26.35 | 0.485 | 25.71 | 0.506 |
| Cat | 30.95 | 0.297 | 25.73 | 0.383 | 22.19 | 0.430 | 20.35 | 0.458 | 24.74 | 0.430 | 21.49 | 0.460 |
| Deer | 33.70 | 0.330 | 32.25 | 0.370 | 18.94 | 0.530 | 18.02 | 0.555 | 22.22 | 0.496 | 18.98 | 0.531 |
| Dog | 31.27 | 0.473 | 27.24 | 0.546 | 22.59 | 0.596 | 22.74 | 0.612 | 29.63 | 0.539 | 25.47 | 0.559 |
| Frog | 39.26 | 0.413 | 30.36 | 0.501 | 19.77 | 0.654 | 16.08 | 0.707 | 20.71 | 0.646 | 19.14 | 0.667 |
| Horse | 26.60 | 0.543 | 20.32 | 0.620 | 16.18 | 0.713 | 14.97 | 0.722 | 17.92 | 0.662 | 15.95 | 0.656 |
| Ship | 26.72 | 0.539 | 19.04 | 0.640 | 13.94 | 0.743 | 13.84 | 0.737 | 15.04 | 0.724 | 16.12 | 0.695 |
| Truck | 30.37 | 0.505 | 20.51 | 0.634 | 15.18 | 0.710 | 13.33 | 0.752 | 15.00 | 0.702 | 16.28 | 0.675 |

Table 7: CG-DLSM

| Class | 5% labeled data intra-FID | Acc | 20% labeled data intra-FID | Acc | 40% labeled data intra-FID | Acc | 60% labeled data intra-FID | Acc | 80% labeled data intra-FID | Acc | 100% labeled data intra-FID | Acc |
|---|---|---|---|---|---|---|---|---|---|---|---|---|
| Airplane | 38.28 | 0.421 | 26.93 | 0.466 | 24.49 | 0.515 | 20.47 | 0.585 | 19.69 | 0.609 | 19.77 | 0.609 |
| Automobile | 30.96 | 0.440 | 28.17 | 0.511 | 23.32 | 0.565 | 19.67 | 0.620 | 17.87 | 0.657 | 16.97 | 0.670 |
| Bird | 34.03 | 0.245 | 36.13 | 0.303 | 33.32 | 0.335 | 26.80 | 0.429 | 24.45 | 0.464 | 25.44 | 0.453 |
| Cat | 42.70 | 0.240 | 31.44 | 0.294 | 29.76 | 0.320 | 26.44 | 0.351 | 24.26 | 0.394 | 23.09 | 0.404 |
| Deer | 36.83 | 0.261 | 33.23 | 0.317 | 30.77 | 0.359 | 23.50 | 0.447 | 22.39 | 0.472 | 20.83 | 0.487 |
| Dog | 39.17 | 0.325 | 36.51 | 0.394 | 34.49 | 0.426 | 29.87 | 0.482 | 28.01 | 0.508 | 28.38 | 0.507 |
| Frog | 43.88 | 0.341 | 35.97 | 0.428 | 32.25 | 0.475 | 23.70 | 0.587 | 20.24 | 0.631 | 21.50 | 0.612 |
| Horse | 47.08 | 0.389 | 27.87 | 0.476 | 26.50 | 0.510 | 21.22 | 0.586 | 20.11 | 0.602 | 19.01 | 0.614 |
| Ship | 38.17 | 0.453 | 28.90 | 0.514 | 27.44 | 0.543 | 19.92 | 0.628 | 20.14 | 0.643 | 18.76 | 0.653 |
| Truck | 33.38 | 0.425 | 32.65 | 0.486 | 30.91 | 0.520 | 23.63 | 0.597 | 20.45 | 0.648 | 22.11 | 0.628 |

Table 8: CG-LS

| Class | 5% labeled data | | 20% labeled data | | 40% labeled data | | 60% labeled data | | 80% labeled data | | 100% labeled data | |
|---|---|---|---|---|---|---|---|---|---|---|---|---|
| | intra-FID | Acc | intra-FID | Acc | intra-FID | Acc | intra-FID | Acc | intra-FID | Acc | intra-FID | Acc |
| Airplane | 25.68 | 0.581 | 21.84 | 0.618 | 15.97 | 0.721 | 16.71 | 0.689 | 20.65 | 0.605 | 16.97 | 0.685 |
| Automobile | 23.91 | 0.552 | 23.17 | 0.582 | 13.64 | 0.726 | 13.52 | 0.742 | 16.83 | 0.694 | 14.79 | 0.726 |
| Bird | 37.44 | 0.290 | 37.86 | 0.324 | 23.09 | 0.497 | 21.56 | 0.516 | 25.78 | 0.473 | 21.72 | 0.534 |
| Cat | 29.78 | 0.305 | 26.32 | 0.394 | 21.77 | 0.433 | 20.71 | 0.454 | 24.02 | 0.425 | 21.28 | 0.439 |
| Deer | 32.02 | 0.353 | 29.87 | 0.360 | 18.83 | 0.545 | 18.48 | 0.540 | 23.44 | 0.488 | 17.63 | 0.568 |
| Dog | 30.47 | 0.484 | 25.72 | 0.564 | 22.81 | 0.592 | 23.42 | 0.587 | 28.91 | 0.523 | 22.22 | 0.600 |
| Frog | 35.59 | 0.444 | 31.11 | 0.510 | 17.79 | 0.688 | 17.99 | 0.680 | 23.07 | 0.613 | 18.95 | 0.682 |
| Horse | 25.50 | 0.555 | 18.66 | 0.646 | 15.70 | 0.716 | 15.50 | 0.704 | 20.11 | 0.615 | 15.99 | 0.687 |
| Ship | 24.41 | 0.572 | 24.08 | 0.588 | 13.61 | 0.749 | 14.47 | 0.742 | 19.35 | 0.679 | 15.31 | 0.728 |
| Truck | 27.57 | 0.527 | 22.91 | 0.636 | 15.67 | 0.716 | 14.87 | 0.725 | 20.86 | 0.650 | 16.19 | 0.712 |

Table 9: CG-JR

| Class | 5% labeled data | | 20% labeled data | | 40% labeled data | | 60% labeled data | | 80% labeled data | | 100% labeled data | |
|---|---|---|---|---|---|---|---|---|---|---|---|---|
| | intra-FID | Acc | intra-FID | Acc | intra-FID | Acc | intra-FID | Acc | intra-FID | Acc | intra-FID | Acc |
| Airplane | 27.09 | 0.570 | 20.21 | 0.659 | 17.60 | 0.697 | 16.95 | 0.675 | 17.85 | 0.657 | 15.70 | 0.704 |
| Automobile | 25.44 | 0.549 | 18.70 | 0.638 | 13.21 | 0.734 | 14.88 | 0.711 | 13.71 | 0.740 | 13.20 | 0.746 |
| Bird | 38.41 | 0.273 | 29.47 | 0.387 | 23.78 | 0.467 | 24.24 | 0.474 | 22.10 | 0.514 | 21.69 | 0.518 |
| Cat | 31.38 | 0.299 | 25.24 | 0.368 | 22.12 | 0.425 | 21.10 | 0.431 | 20.67 | 0.435 | 19.94 | 0.456 |
| Deer | 33.47 | 0.331 | 25.95 | 0.421 | 20.10 | 0.524 | 20.97 | 0.491 | 18.47 | 0.546 | 17.64 | 0.562 |
| Dog | 30.76 | 0.483 | 25.28 | 0.549 | 22.70 | 0.593 | 22.64 | 0.597 | 23.90 | 0.569 | 22.87 | 0.592 |
| Frog | 39.77 | 0.413 | 27.06 | 0.546 | 20.11 | 0.646 | 19.83 | 0.649 | 18.81 | 0.662 | 17.82 | 0.683 |
| Horse | 25.58 | 0.561 | 19.47 | 0.649 | 16.34 | 0.726 | 16.61 | 0.675 | 16.82 | 0.673 | 14.74 | 0.708 |
| Ship | 25.52 | 0.547 | 17.91 | 0.677 | 14.39 | 0.736 | 15.96 | 0.706 | 14.39 | 0.732 | 14.04 | 0.735 |
| Truck | 28.50 | 0.518 | 20.99 | 0.628 | 15.99 | 0.706 | 17.27 | 0.682 | 16.92 | 0.695 | 14.73 | 0.724 |

Table 10: Cond

| Class | 5% labeled data | | 20% labeled data | | 40% labeled data | | 60% labeled data | | 80% labeled data | | 100% labeled data | |
|---|---|---|---|---|---|---|---|---|---|---|---|---|
| | intra-FID | Acc | intra-FID | Acc | intra-FID | Acc | intra-FID | Acc | intra-FID | Acc | intra-FID | Acc |
| Airplane | 19.95 | 0.805 | 14.88 | 0.823 | 14.66 | 0.858 | 12.72 | 0.852 | 12.75 | 0.809 | 11.98 | 0.858 |
| Automobile | 13.15 | 0.758 | 9.37 | 0.846 | 9.20 | 0.907 | 8.79 | 0.916 | 8.47 | 0.881 | 8.29 | 0.921 |
| Bird | 25.12 | 0.479 | 16.68 | 0.647 | 14.45 | 0.716 | 14.99 | 0.715 | 15.76 | 0.697 | 14.30 | 0.755 |
| Cat | 22.95 | 0.448 | 16.93 | 0.566 | 15.78 | 0.590 | 15.31 | 0.617 | 15.49 | 0.653 | 14.91 | 0.670 |
| Deer | 20.10 | 0.532 | 13.50 | 0.675 | 11.61 | 0.763 | 11.47 | 0.761 | 11.36 | 0.729 | 10.74 | 0.779 |
| Dog | 23.03 | 0.648 | 19.39 | 0.694 | 16.08 | 0.780 | 16.19 | 0.763 | 17.67 | 0.755 | 16.32 | 0.781 |
| Frog | 19.12 | 0.721 | 13.80 | 0.787 | 13.06 | 0.839 | 12.64 | 0.857 | 13.05 | 0.838 | 12.00 | 0.899 |
| Horse | 19.05 | 0.815 | 14.56 | 0.821 | 13.51 | 0.887 | 12.25 | 0.869 | 11.41 | 0.822 | 11.09 | 0.870 |
| Ship | 13.34 | 0.798 | 10.85 | 0.836 | 9.97 | 0.887 | 9.62 | 0.869 | 9.75 | 0.855 | 8.99 | 0.890 |
| Truck | 13.68 | 0.752 | 9.78 | 0.826 | 8.32 | 0.859 | 8.21 | 0.878 | 9.00 | 0.844 | 8.36 | 0.871 |

Table 11: CFG-labeled

| Class | 5% labeled data | | 20% labeled data | | 40% labeled data | | 60% labeled data | | 80% labeled data | | 100% labeled data | |
|---|---|---|---|---|---|---|---|---|---|---|---|---|
| | intra-FID | Acc | intra-FID | Acc | intra-FID | Acc | intra-FID | Acc | intra-FID | Acc | intra-FID | Acc |
| Airplane | 46.22 | 0.961 | 33.58 | 0.877 | 16.37 | 0.931 | 12.24 | 0.965 | 11.80 | 0.970 | 11.26 | 0.963 |
| Automobile | 30.55 | 0.989 | 19.50 | 0.968 | 11.17 | 0.988 | 9.16 | 0.992 | 8.83 | 0.994 | 8.59 | 0.992 |
| Bird | 57.41 | 0.930 | 37.85 | 0.847 | 15.50 | 0.912 | 12.19 | 0.941 | 11.79 | 0.950 | 11.53 | 0.954 |
| Cat | 64.70 | 0.917 | 46.44 | 0.861 | 18.16 | 0.900 | 13.85 | 0.911 | 13.74 | 0.930 | 13.24 | 0.926 |
| Deer | 46.66 | 0.959 | 29.04 | 0.846 | 11.95 | 0.938 | 10.49 | 0.957 | 10.52 | 0.969 | 9.87 | 0.967 |
| Dog | 51.47 | 0.876 | 42.24 | 0.849 | 16.16 | 0.930 | 12.86 | 0.956 | 12.31 | 0.954 | 12.22 | 0.959 |
| Frog | 52.77 | 0.946 | 44.03 | 0.920 | 15.54 | 0.978 | 13.41 | 0.990 | 11.79 | 0.994 | 11.77 | 0.994 |
| Horse | 37.29 | 0.956 | 25.04 | 0.914 | 13.67 | 0.964 | 12.11 | 0.976 | 10.61 | 0.984 | 11.40 | 0.990 |
| Ship | 36.18 | 0.995 | 28.75 | 0.952 | 11.45 | 0.972 | 8.85 | 0.983 | 8.60 | 0.986 | 8.31 | 0.983 |
| Truck | 27.44 | 0.965 | 20.18 | 0.897 | 9.29 | 0.966 | 7.60 | 0.985 | 7.51 | 0.983 | 7.58 | 0.984 |

Table 12: CFG-all

| Class | 5% labeled data | | 20% labeled data | | 40% labeled data | | 60% labeled data | | 80% labeled data | | 100% labeled data | |
|---|---|---|---|---|---|---|---|---|---|---|---|---|
| | intra-FID | Acc | intra-FID | Acc | intra-FID | Acc | intra-FID | Acc | intra-FID | Acc | intra-FID | Acc |
| Airplane | 45.06 | 0.982 | 31.93 | 0.918 | 15.57 | 0.958 | 12.64 | 0.964 | 11.76 | 0.967 | 11.26 | 0.963 |
| Automobile | 29.94 | 0.993 | 18.53 | 0.992 | 11.54 | 0.997 | 9.99 | 0.995 | 8.48 | 0.992 | 8.59 | 0.992 |
| Bird | 55.92 | 0.930 | 36.37 | 0.880 | 14.07 | 0.952 | 12.06 | 0.951 | 11.90 | 0.951 | 11.53 | 0.954 |
| Cat | 69.53 | 0.962 | 44.56 | 0.918 | 16.92 | 0.930 | 14.34 | 0.936 | 13.51 | 0.922 | 13.24 | 0.926 |
| Deer | 56.38 | 0.970 | 28.75 | 0.907 | 12.50 | 0.966 | 10.13 | 0.961 | 10.25 | 0.972 | 9.87 | 0.967 |
| Dog | 52.48 | 0.907 | 39.27 | 0.884 | 15.23 | 0.962 | 12.88 | 0.959 | 12.44 | 0.953 | 12.22 | 0.959 |
| Frog | 64.09 | 0.966 | 45.12 | 0.960 | 14.54 | 0.990 | 12.54 | 0.992 | 12.89 | 0.993 | 11.77 | 0.994 |
| Horse | 37.65 | 0.946 | 24.21 | 0.950 | 13.95 | 0.981 | 12.38 | 0.983 | 12.40 | 0.982 | 11.40 | 0.990 |
| Ship | 35.75 | 0.998 | 24.28 | 0.977 | 10.98 | 0.983 | 9.03 | 0.983 | 8.42 | 0.985 | 8.31 | 0.983 |
| Truck | 26.52 | 0.985 | 19.39 | 0.971 | 8.96 | 0.981 | 7.81 | 0.988 | 7.33 | 0.984 | 7.58 | 0.984 |

# B MORE DETAILED INTRODUCTION ON SCORE-BASED GENERATIVE MODELING THROUGH SDE

## B.1 LEARNING THE SCORE FUNCTION

When learning the score function, the goal is to choose the best function from a family of functions $\{s(\boldsymbol{x}; \boldsymbol{\theta})\}_{\boldsymbol{\theta}}$, such as deep learning models parameterized by $\boldsymbol{\theta}$, to approximate the score function $\nabla_{\boldsymbol{x}} \log p(\boldsymbol{x})$ of interest. Learning is based on data $\{\boldsymbol{x}_n\}_{n=1}^{N}$ assumed to be sampled from $p(\boldsymbol{x})$. It has been shown that this can be achieved by optimizing the in-sample version of the following score-matching loss over $\theta$:

$$\mathcal{L}_{\text{SM}} = \mathbb{E}_{p(\boldsymbol{x})} \left[ tr(\nabla_{\boldsymbol{x}} s(\boldsymbol{x}; \boldsymbol{\theta})) + \frac{1}{2} \|s(\boldsymbol{x}; \boldsymbol{\theta})\|_2^2 \right],$$

where $tr(\cdot)$ denotes the trace of a matrix and $\nabla_{\boldsymbol{x}} s(\boldsymbol{x}; \boldsymbol{\theta}) = \nabla_{\boldsymbol{x}}^2 \log p(x)$ is the Hessian matrix of log-likelihood $\log p(\boldsymbol{x})$. Calculating the score-matching loss requires $O(d)$ computation passes for $\boldsymbol{x} \in \mathbb{R}^d$, which makes the optimization process computationally prohibitive on high-dimensional data.

Several attempts (Kingma & Cun, 2010; Martens et al., 2012; Vincent, 2011; Song et al., 2019) have been made to address these computational challenges by approximating or transforming score matching into equivalent objectives. One current standard approach is called denoise score matching (DSM) (Vincent, 2011), which instead learns the score function of a noise-perturbed data distribution $q(\tilde{\boldsymbol{x}})$. DSM typically assumes that $q(\tilde{\boldsymbol{x}})$ comes from the original distribution $p(\boldsymbol{x})$ injected with a pre-specified noise $q(\tilde{\boldsymbol{x}}|\boldsymbol{x})$. It has been proved (Vincent, 2011) that the score function can be learned by minimizing the in-sample version of

$$\mathbb{E}_{q(\tilde{\boldsymbol{x}}|\boldsymbol{x})p(\boldsymbol{x})} \left[ \frac{1}{2} \|s(\tilde{\boldsymbol{x}}; \boldsymbol{\theta}) - \nabla_{\tilde{\boldsymbol{x}}} \log q(\tilde{\boldsymbol{x}}|\boldsymbol{x})\|_2^2 \right],$$

where $\nabla_{\tilde{\boldsymbol{x}}} \log q(\tilde{\boldsymbol{x}}|\boldsymbol{x})$ is the score function of the noise distribution centered at $\boldsymbol{x}$. DSM is generally more efficient than the original score matching and is scalable to high-dimensional data as it replaces the heavy computation on the Hessian matrix with simple perturbations that can be efficiently computed from data.

## B.2 GENERATION FROM THE SCORE FUNCTION BY DIFFUSION

Assume that we seek to sample from some unknown target distribution $p(\boldsymbol{x}) = p_0(\boldsymbol{x})$, and the distribution can be diffused to a known prior distribution $p_T(\boldsymbol{x})$ through a Markov chain that is described with a stochastic differential equation (SDE) (Song et al., 2021): $d\boldsymbol{x} = f(\boldsymbol{x}, t)dt + g(t)d\boldsymbol{w}$, where the Markov chain is computed for $0 \leq t < T$ using the drift function $f(\boldsymbol{x}, t)$ that describes the overall movement and the dispersion function $g(t)$ that describes how the noise $\boldsymbol{w}$ from a standard Wiener process enters the system.

To sample from $p(\boldsymbol{x}) = p_0(\boldsymbol{x})$, Song et al. (2021) proposes to reverse the SDE from $p_T(\boldsymbol{x})$ to $p_0(\boldsymbol{x})$, which turns out to operate with another SDE (Eq. 1). Given the score function $s(\boldsymbol{x}, t)$, the diffusion process in Eq. 1 can then be used to take any data sampled from the known $p_T(\boldsymbol{x})$ to a sample from the unknown $p(\boldsymbol{x}) = p_0(\boldsymbol{x})$.

The time-dependent score function $s(\boldsymbol{x}, t; \boldsymbol{\theta})$ can be learned by minimizing a time-generalized (in-sample) version of the DSM loss because the diffusion process can be viewed as one particular way of injecting noise. The extended DSM loss is defined as

$$\mathcal{L}_{DSM}(\boldsymbol{\theta}) = \mathbb{E}_t \left[ \lambda(t) \mathbb{E}_{\boldsymbol{x}_t, \boldsymbol{x}_0} \left[ \frac{1}{2} \|s(\boldsymbol{x}_t, t; \boldsymbol{\theta}) - s_t(\boldsymbol{x}_t|\boldsymbol{x}_0)\|_2^2 \right] \right],$$

where $t$ is selected uniformly between 0 and $T$, $\boldsymbol{x}_t \sim p_t(\boldsymbol{x})$, $\boldsymbol{x}_0 \sim p_0(\boldsymbol{x})$, $s_t(\boldsymbol{x}_t|\boldsymbol{x}_0)$ denotes the score function of $p_t(\boldsymbol{x}_t|\boldsymbol{x}_0)$, and $\lambda(t)$ is a weighting function that balances the loss of different timesteps.

## C  TRAINING ALGORITHM FOR SEMI-SUPERVISED SELF-CALIBRATING CLASSIFIER

---

**Algorithm 1** Semi-supervised classifier training with self-calibration loss

---

**Input:** Labeled data $D_l$, unlabeled data $D_u$
Initialize the time-dependent classifier $f(\boldsymbol{x}, y, t; \boldsymbol{\phi})$ randomly
**repeat**
  Sample data $(\boldsymbol{x}_l, y_l) \sim D_l$, $\boldsymbol{x}_u \sim D_u$
  Sample timesteps $t_l, t_u \sim \text{Uniform}(1, T)$
  Obtain perturbed data $\tilde{\boldsymbol{x}}_l \sim p_{t_l}(\boldsymbol{x}|\boldsymbol{x}_l)$, $\tilde{\boldsymbol{x}}_u \sim p_{t_u}(\boldsymbol{x}|\boldsymbol{x}_u)$
  Calculate $\mathcal{L}_{CE} = \text{CrossEntropy}(f(\boldsymbol{x}_l, y, t; \boldsymbol{\phi}), y_l)$
  Calculate $\mathcal{L}_{SC} = \mathbb{E}_{(\boldsymbol{x},t) \in \{(\boldsymbol{x}_l, t_l),(\boldsymbol{x}_u, t_u)\}} \left[ \frac{1}{2}\lambda(t)\|\nabla_{\boldsymbol{x}} \log \Sigma_y \exp(f(\boldsymbol{x}, y, t; \boldsymbol{\phi})) - s_t(\boldsymbol{x}_t|\boldsymbol{x}_0)\|_2^2 \right]$

  Take gradient step on $\mathcal{L}_{CLS} = \mathcal{L}_{CE} + \mathcal{L}_{SC}$
**until** converged

---

## D  ADDITIONAL EXPERIMENTAL DETAILS

We followed NCSN++ (Song et al., 2021) to implement the unconditional score estimation model. We also adapted the encoder part of NCSN++ as the classifier used in CGSGM (Dhariwal & Nichol, 2021) and its variants, e.g., CG-DLSM or the proposed CG-SC. For the sampling method, we used Predictor-Corrector (PC) samplers (Song et al., 2021) with 1000 sampling steps. The SDE was selected as the VE-SDE framework proposed by Song et al. (2021). The hyper-parameter introduced in Eq. 8 is tuned between $\{10, 1, 0.1, 0.01\}$ for fully-supervised settings, and selected to be 1 in semi-supervised settings due to limited computational resources. The scaling factor $\lambda_{CG}$ introduced in Eq. 4 is tuned within $\{0.5, 0.8, 1.0, 1.2, 1.5, 2.0, 2.5\}$ to obtain the best intra-FID. A similar scaling factor $\lambda_{CFG}$ for classifier-free SGMs is tuned within $\{0, 0.1, 0.2, 0.4\}$ to obtain the best intra-FID. The balancing factors of the DLSM loss and the Jacobian regularization loss are selected to be 1 and 0.01, respectively, as suggested in the original papers. The smoothing factor of label-smoothing is tuned between $\{0.1, 0.05\}$ for the better intra-FID.

## E  ATTEMPTS TO INCORPORATE JEM INTO THE TIME-DEPENDENT FRAMEWORK

There are two key differences between the original JEM framework and the time-dependent framework. First, the classifier has been augmented to take timestep $t$ as an additional input. Second, the MCMC sampling component of the JEM training algorithm has been modified to fit in the time-dependent framework.

The adaptation for the first difference is intuitive, having been addressed when we transitioned from standard classifiers to those that are time-dependent. For the second difference, when initially integrating MCMC sampling with the time-dependent framework, we employed a methodology that uniformly sampled timesteps and directly incorporated the remaining parts of the JEM training algorithm. However, we observed early divergence in the training loss during multiple runs, both with the CIFAR-10 and the toy dataset. A pattern emerged: the MCMC algorithm often produced extremely high-energy samples. The loss and magnitude of backpropagated gradients resulting from this corrupted sample became substantially larger than other samples. This often caused the training loss to diverge. Consequently, we made the following refinements:

1. **Sample Initialization:** Instead of uniformly initializing samples between 0 and 1 without considering the corresponding timestep, we opted to diffuse those samples to the corresponding timestep using the forward SDE to bring the initialized samples closer to the perturbed data distributions at different timesteps.

2. **Step Size:** Rather than employing a consistent step size across all timesteps, we adopted the step-size-selection strategy proposed for annealed Langevin dynamics (Song & Ermon,

2019), where step sizes are selected to be directly proportional to the variance of the injected noise. Note that the ratio of step-size to noise was retained from the original JEM framework.

3. **Short-run MCMC:** Instead of using PCD with 20 steps, we adopted short-run MCMC with 100 steps. This change, as suggested in Grathwohl et al. (2020), is known to improve the stability of JEM training.

Following these refinements, we observed enhanced stability in training. However, we still observed occasional divergences in training loss. Nevertheless, successful training of a time-dependent classifier was achieved on the toy dataset, as shown in Section 3.3), but the expected timeframe for incorporating relatively stable JEM training into the CIFAR-10 dataset stands at approximately 50 days. This cost is prohibitive, especially when considering the additional fine-tuning required to obtain hyperparameters that are reasonably close to the optimal configuration.

## F    QUANTITATIVE MEASUREMENTS OF TOY DATASET

Table 13: Mean squared error (MSE) and cosine similarity (CS) of all CGSGM methods tested on the toy dataset

| Method | Gradient MSE ($\downarrow$) | Gradient CS ($\uparrow$) | Cond-Score CS ($\uparrow$) |
|--------|-----------------------------|--------------------------|----------------------------|
| CG | 8.7664 | 0.3265 | 0.9175 |
| CG + scaling | 8.1916 | 0.3348 | 0.9447 |
| CG-SC | 7.1558 | 0.5667 | 0.9454 |
| CG-SC + scaling | **5.6376** | 0.5758 | 0.9689 |
| CG-DLSM | 8.1183 | 0.4450 | 0.9316 |
| CG-DLSM + scaling | 8.0671 | 0.4450 | 0.9328 |
| CG-JEM | 8.5577 | 0.6422 | 0.9670 |
| CG-JEM + scaling | 8.5577 | **0.6429** | **0.9709** |

Table 13 shows the quantitative measurements of the methods on the toy dataset. First, we compared the gradients $\nabla_{\boldsymbol{x}} \log p(y|\boldsymbol{x})$ estimated by the classifiers with the ground truth by calculating the mean squared error (first column) and cosine similarity (second column). The results were calculated by averaging over all $(x, y) \in \{(x, y)|x \in \{-12, -11.5, \dots, 11.5, 12\}, y \in \{-8, -7.5, \dots, 7.5, 8.0\}\}$. We observe that after self-calibration, the mean squared error of the estimated gradients is $18\%$ lower; tuning the scaling factor further improves this to $36\%$. This improvement after scaling implies that the direction of gradients better aligns with the ground truth, and scaling further reduces the mismatch between the magnitude of the classifier and the ground truth. In terms of cosine similarity, self-calibration grants the classifiers an improvement of $42\%$. The numerical results agree with our previous observation that after self-calibration, classifiers better align with the ground truth in terms of both direction and magnitude.

Then, we add the unconditional score of the training data distribution to the classifier gradients to calculate the conditional scores and compare the results with the ground truth. The resulting classifiers estimate conditional scores with a cosine similarity of $0.9175$ even without self-calibration. This shows that with a well-trained unconditional SGM—in this case, where we use the ground-truth unconditional score—CGSGM is able to produce conditional scores pointing in the correct directions in most cases. This explains why the original CGSGM generates samples with decent quality. After applying self-calibration loss and the scaling method, we further improve the cosine similarity to $0.9689$, which we believe enhances the quality of class-conditional generation.

## G    CLASSIFIER-ONLY GENERATION BY INTERPRETING CLASSIFIERS AS SGMS

In this section, we show the results when taking the score estimated by a classifier as an unconditional SGM. For unconditional generation, the classifier is used to estimate the unconditional score; for conditional generation, both terms in Eq. 3 are estimated by classifiers. In other words, the

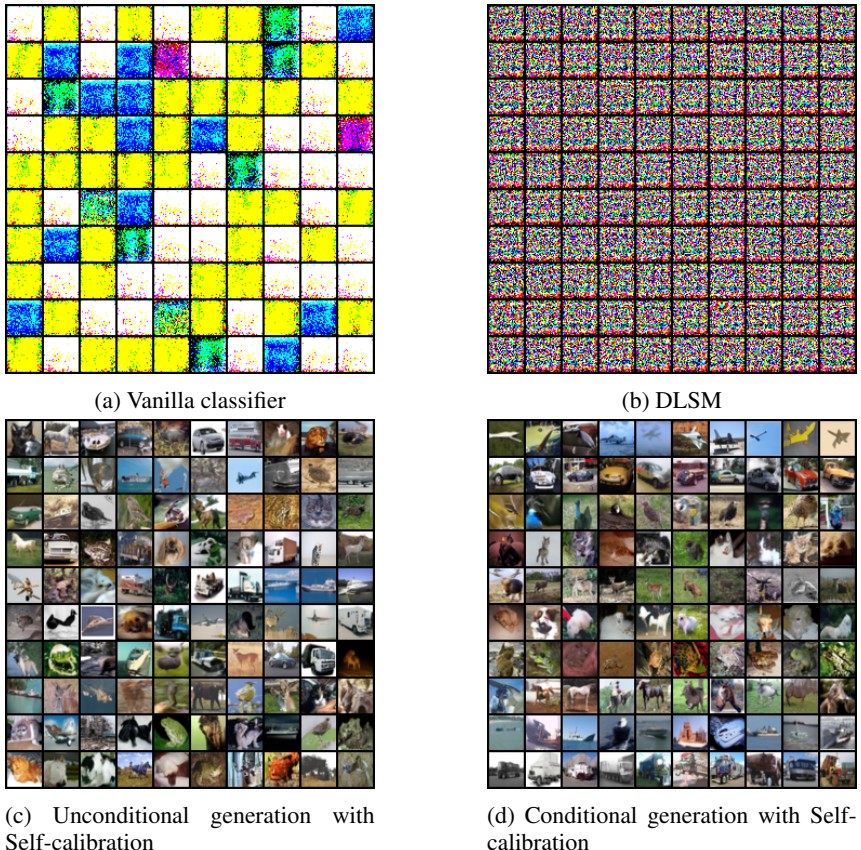

(a) Vanilla classifier

(b) DLSM

(c) Unconditional generation with Self-calibration

(d) Conditional generation with Self-calibration

Figure 5: Generated images from classifier-only score estimation

time-dependent unconditional score $\nabla_{\boldsymbol{x}} \log p_t(\boldsymbol{x})$ can be written as

$$\nabla_{\boldsymbol{x}} \log p_t(\boldsymbol{x}) = \nabla_{\boldsymbol{x}} \log \Sigma_y \exp f(\boldsymbol{x}, y, t; \boldsymbol{\phi}) \tag{9}$$

where $f(\boldsymbol{x}, y, t; \boldsymbol{\phi})$ is the logits of the classifier. By adding the gradient of classifier $\nabla_{\boldsymbol{x}} \log p_t(y|\boldsymbol{x})$ to Eq. 9, we obtain the conditional score estimated by a classifier:

$$
\begin{aligned}
\nabla_{\boldsymbol{x}} \log p_t(\boldsymbol{x}|y) &= \nabla_{\boldsymbol{x}} \log p_t(\boldsymbol{x}) + \nabla_{\boldsymbol{x}} \log p_t(y|\boldsymbol{x}) \\
&= \nabla_{\boldsymbol{x}} \log \Sigma_y \exp f(\boldsymbol{x}, y, t; \boldsymbol{\phi}) + \nabla_{\boldsymbol{x}} \log \frac{\exp\left(f(\boldsymbol{x}, y, t; \boldsymbol{\phi})\right)}{\sum_y \exp\left(f(\boldsymbol{x}, y, t; \boldsymbol{\phi})\right)} \\
&= \nabla_{\boldsymbol{x}} \log \Sigma_y \exp f(\boldsymbol{x}, y, t; \boldsymbol{\phi}) + \nabla_{\boldsymbol{x}} f(\boldsymbol{x}, y, t; \boldsymbol{\phi}) - \nabla_{\boldsymbol{x}} \log \Sigma_y \exp f(\boldsymbol{x}, y, t; \boldsymbol{\phi}) \\
&= \nabla_{\boldsymbol{x}} f(\boldsymbol{x}, y, t; \boldsymbol{\phi})
\end{aligned}
$$

Here, the conditional score is essentially the gradient of the logits. Therefore, we sample from $\nabla_{\boldsymbol{x}} \text{LogSumExp}_y f(\boldsymbol{x}, y, t; \boldsymbol{\phi})$ for unconditional generation and $\nabla_{\boldsymbol{x}} f(\boldsymbol{x}, y, t; \boldsymbol{\phi})$ for conditional generation.

Table 14: Quantitative measurements of classifier-only generation with classifier trained using self-calibration loss as regularization

| Method | FID ($\downarrow$) | IS ($\uparrow$) | intra-FID ($\downarrow$) | Acc ($\uparrow$) |
|---|---|---|---|---|
| $\nabla_{\boldsymbol{x}} \log \Sigma_y \exp f(\boldsymbol{x}, y, t; \boldsymbol{\phi})$ | 7.54 | 8.93 | | |
| $\nabla_{\boldsymbol{x}} f(\boldsymbol{x}, y, t; \boldsymbol{\phi})$ | 7.26 | 8.93 | 18.86 | 0.890 |

Without self-calibration, both the vanilla classifier (Fig. 5a) and DLSM (Fig. 5b) are unable to generate meaningful images when interpreted as conditional SGMs; this also occurs for unconditional

generation. This shows that without related regularization, the interpretation of classifiers as SGMs is not naturally learned through the classification task. After adopting self-calibration loss as regularization, Figures 5c and 5d show that not only does $\nabla_{\boldsymbol{x}}\text{LogSumExp}_y f(\boldsymbol{x}, y, t; \boldsymbol{\phi})$ become a more accurate estimator of unconditional score through direct training, $\nabla_{\boldsymbol{x}} f(\boldsymbol{x}, y, t; \boldsymbol{\phi})$ also becomes a better estimator of the conditional score as a side effect. Here, we also include the quantitative measurements of unconditional and conditional classifier-only generation in Table 14.

## H    TUNING THE SCALING FACTOR FOR CLASSIFIER GUIDANCE

This section includes the results when tuning the scaling factor $\lambda_{\text{CG}}$ for classifier guidance with and without self-calibration under the fully-supervised setting.

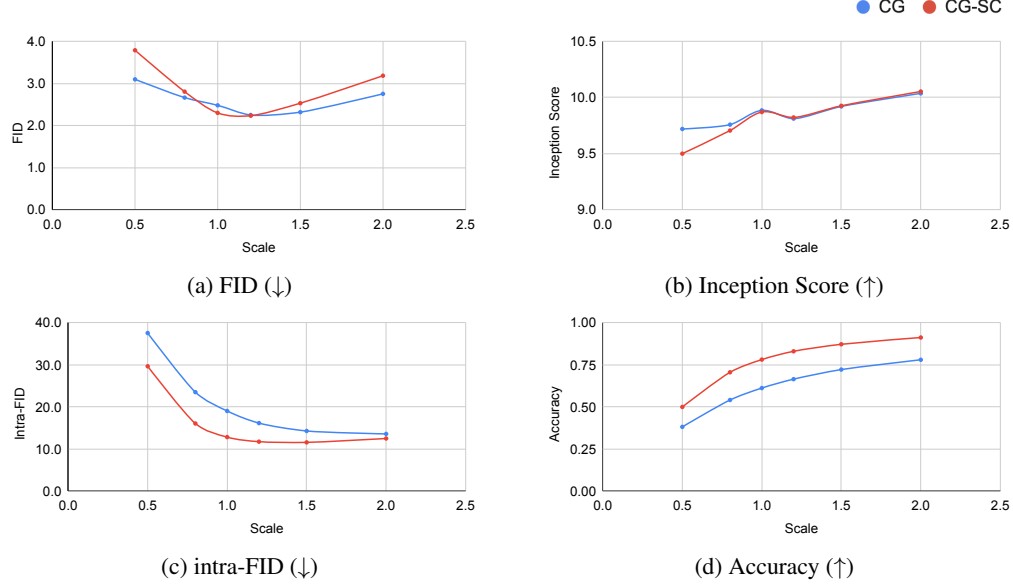

(a) FID ($\downarrow$)  (b) Inception Score ($\uparrow$)

(c) intra-FID ($\downarrow$)  (d) Accuracy ($\uparrow$)

Figure 6: Results when tuning scaling factor $\lambda_{\text{CG}}$ for **CGSGM** (blue, without self-calibration) and **CGSGM-SC** (red, with self-calibration). (a) FID vs. $\lambda_{\text{CG}}$. (b) Inception score vs. $\lambda_{\text{CG}}$. (c) Intra-FID vs. $\lambda_{\text{CG}}$. (d) Generation accuracy vs. $\lambda_{\text{CG}}$. Unconditional metrics (FID and IS) differ little, but we observe a distinct performance gap when evaluated conditionally (intra-FID and accuracy).

Figure 6 shows the result when tuning the scaling factor $\lambda_{\text{CG}}$ for classifier guidance. When tuning $\lambda_{\text{CG}}$ with and without self-calibration, self-calibration has little affect on unconditional performance. However, when evaluated with conditional metrics, the improvement after incorporating self-calibration becomes more significant. The improvement in intra-FID reaches 7.9 whereas generation accuracy improves by as much as 13%.

# I IMAGES GENERATED BY CLASSIFIER GUIDANCE WITH AND WITHOUT SELF-CALIBRATION

This section includes images generated by classifier guidance with (first 6 images) and without (last 6 images) self-calibration after training on various percentages of labeled data. Each row corresponds to a class in the CIFAR-10 dataset. Generated images of all method can be found in the supplementary material.

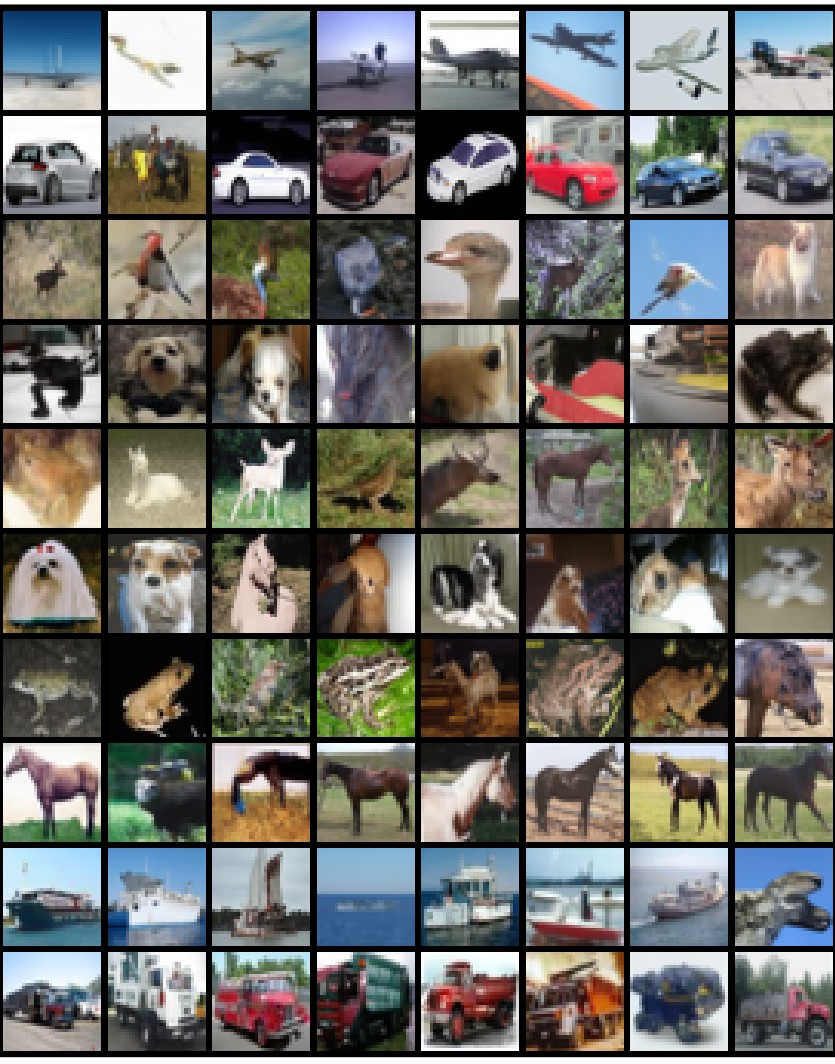

Figure 7: Randomly selected images of classifier guidance with self-calibration (5% labeled data)

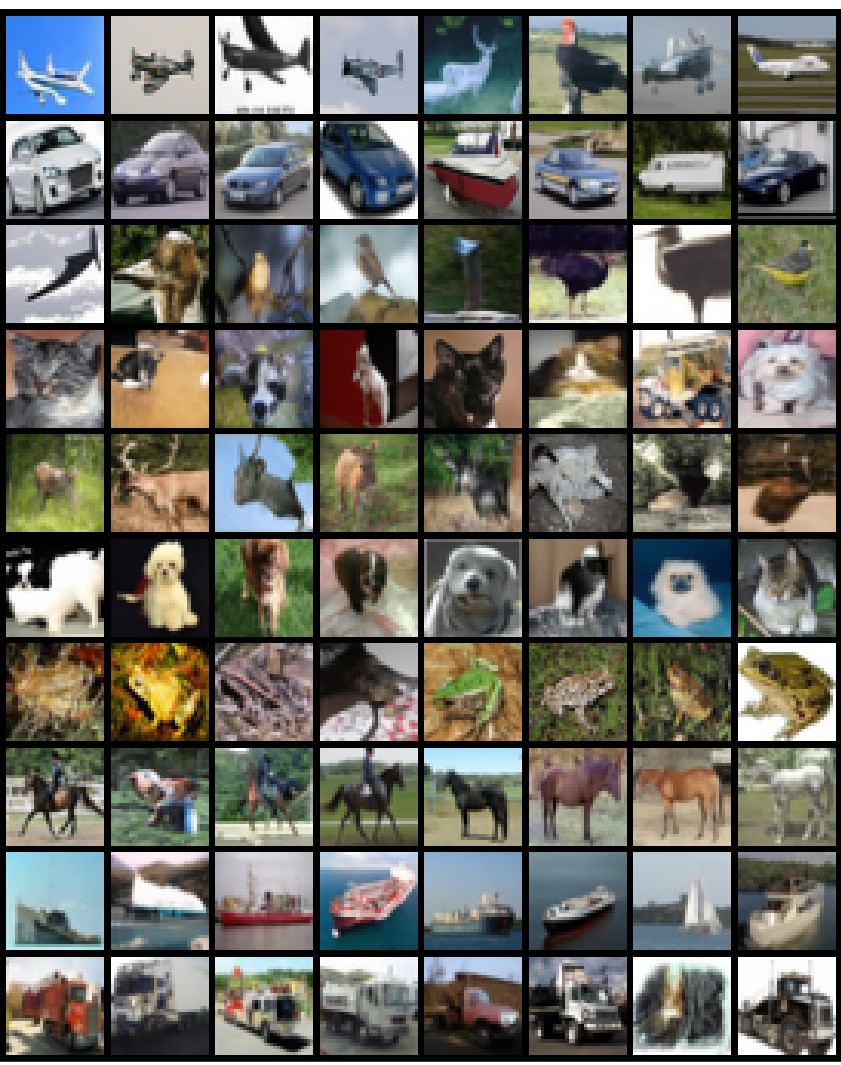

Figure 8: Randomly selected images of classifier guidance with self-calibration (20% labeled data)

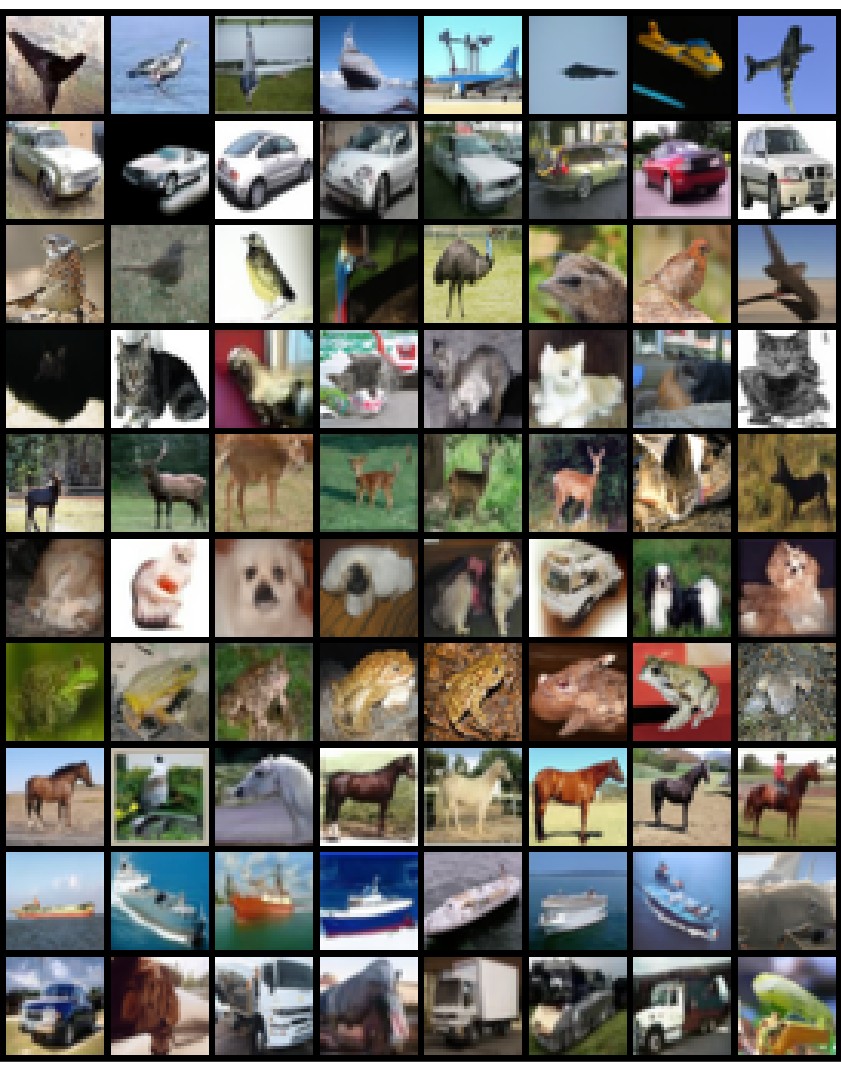

Figure 9: Randomly selected images of classifier guidance with self-calibration (40% labeled data)

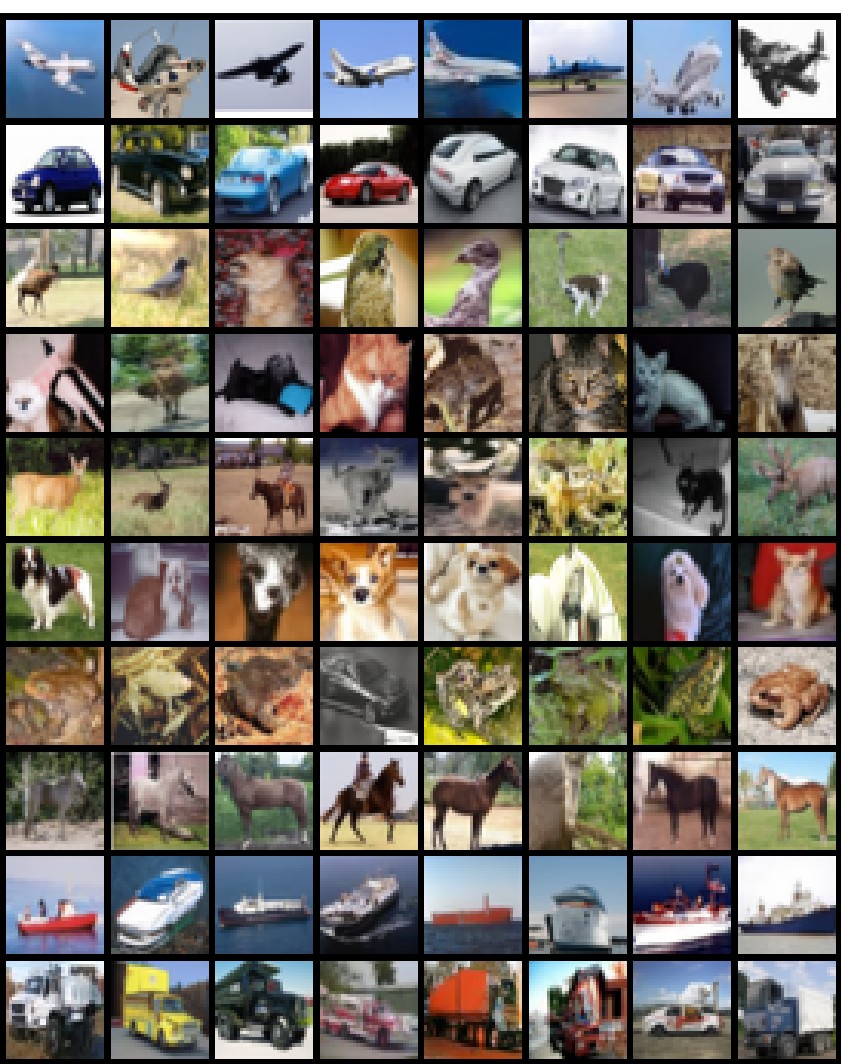

Figure 10: Randomly selected images of classifier guidance with self-calibration (60% labeled data)

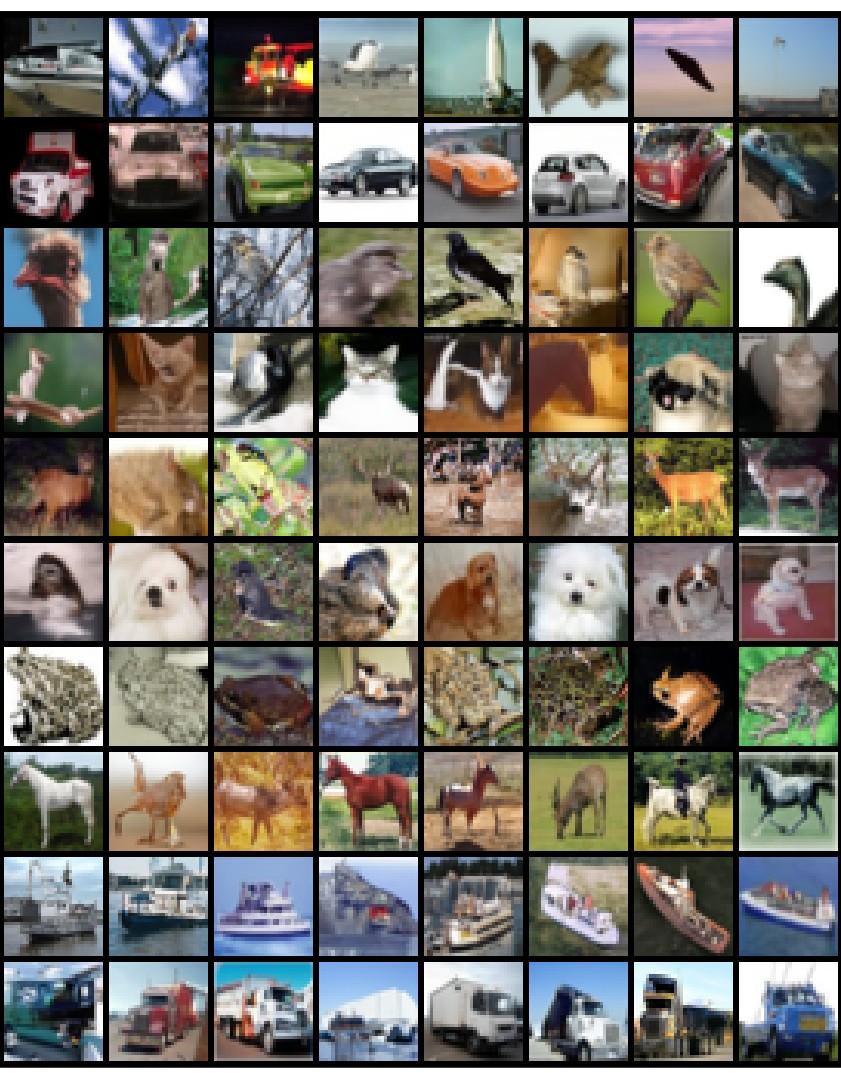

Figure 11: Randomly selected images of classifier guidance with self-calibration (80% labeled data)

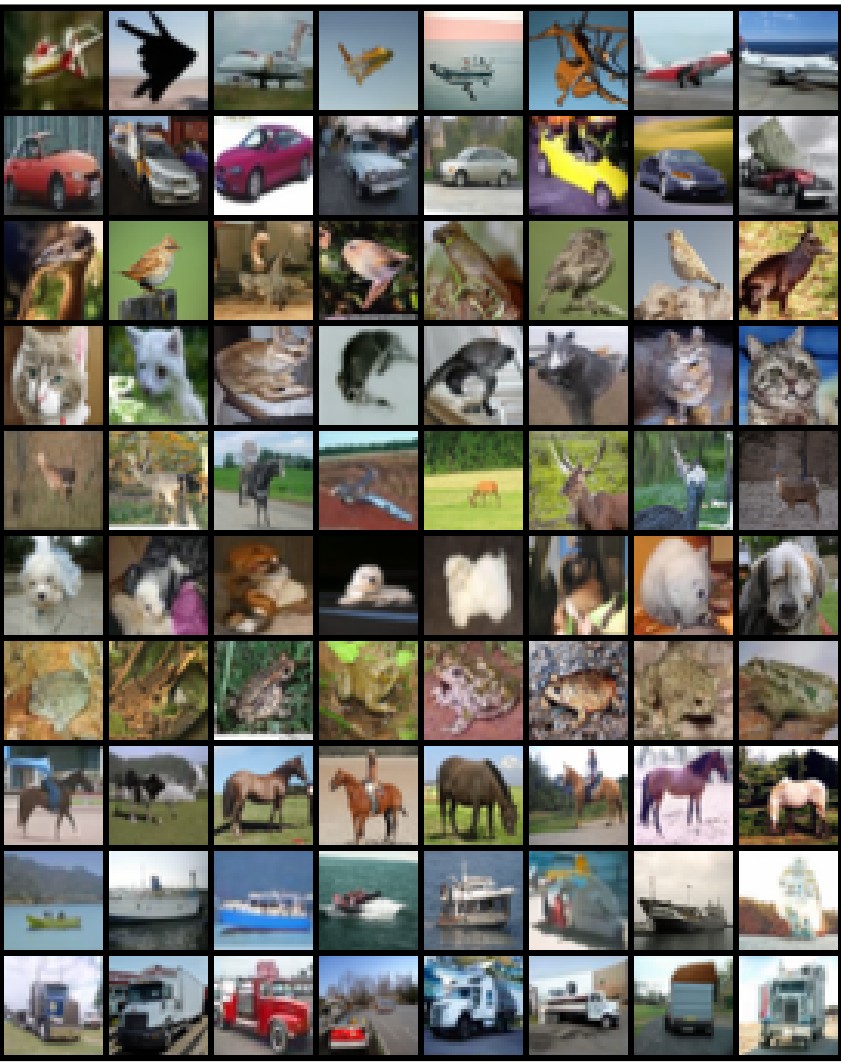

Figure 12: Randomly selected images of classifier guidance with self-calibration (100% labeled data)

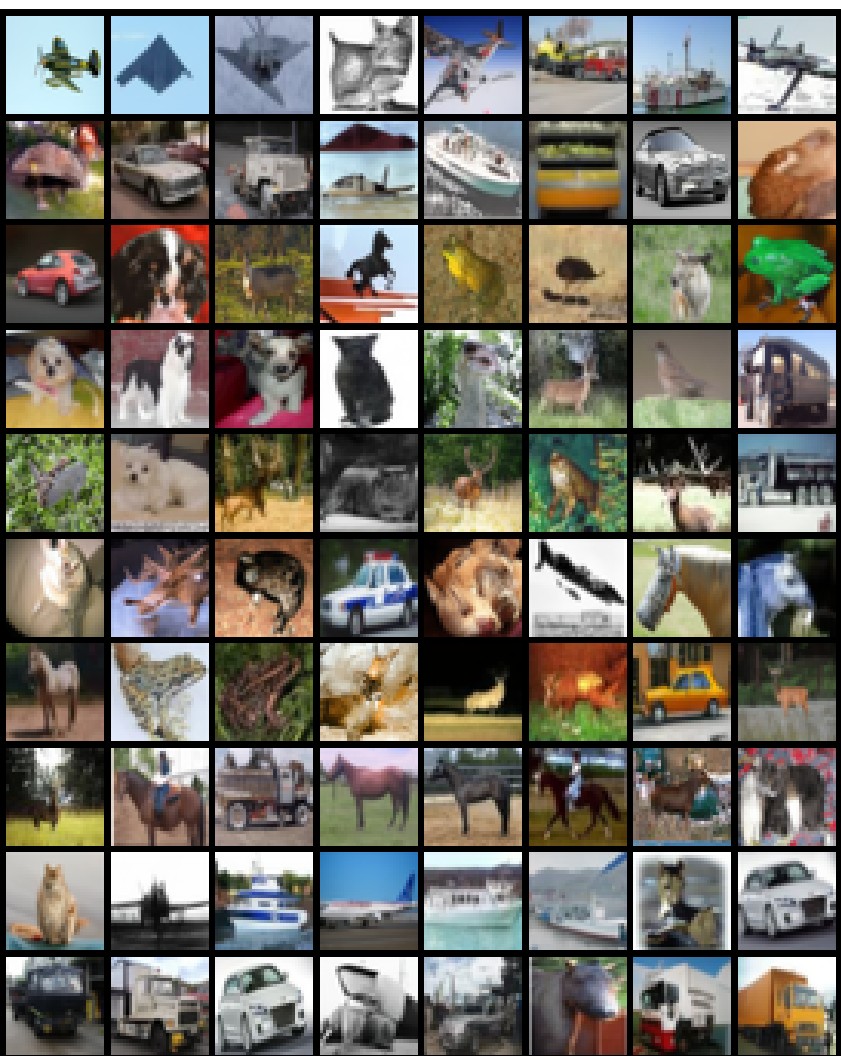

Figure 13: Randomly selected images of vanilla classifier guidance (5% labeled data)

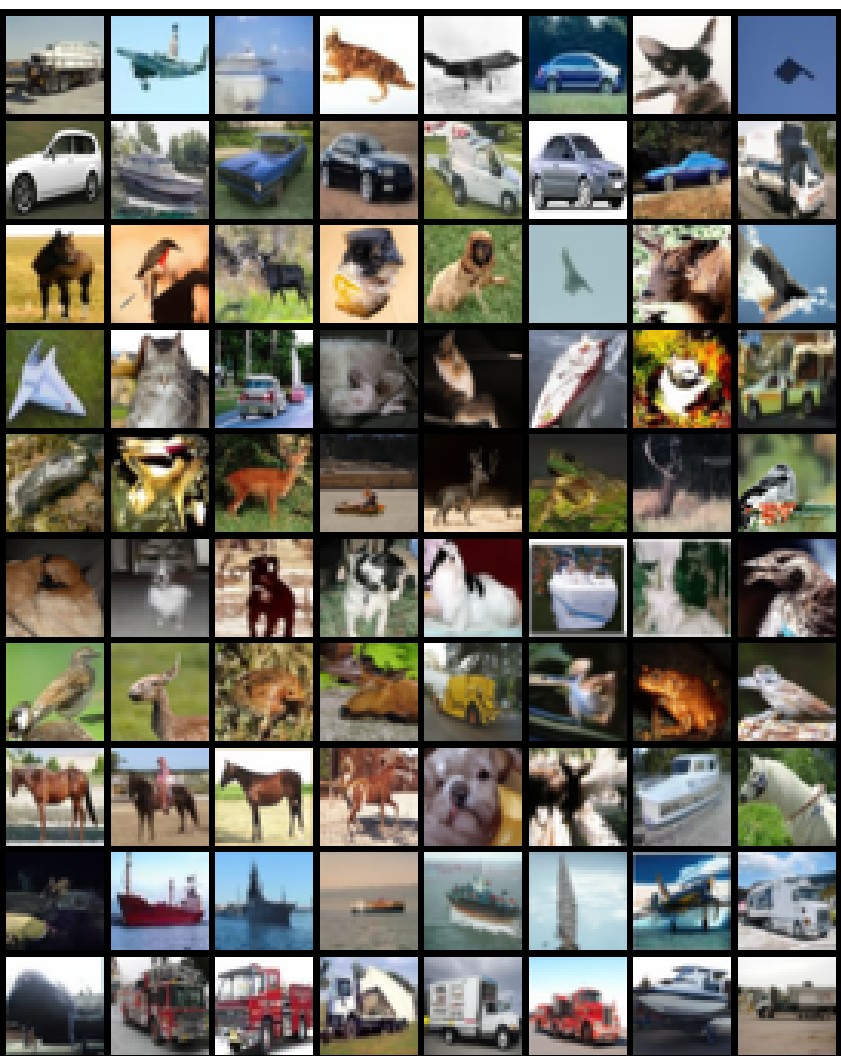

Figure 14: Randomly selected images of vanilla classifier guidance (20% labeled data)

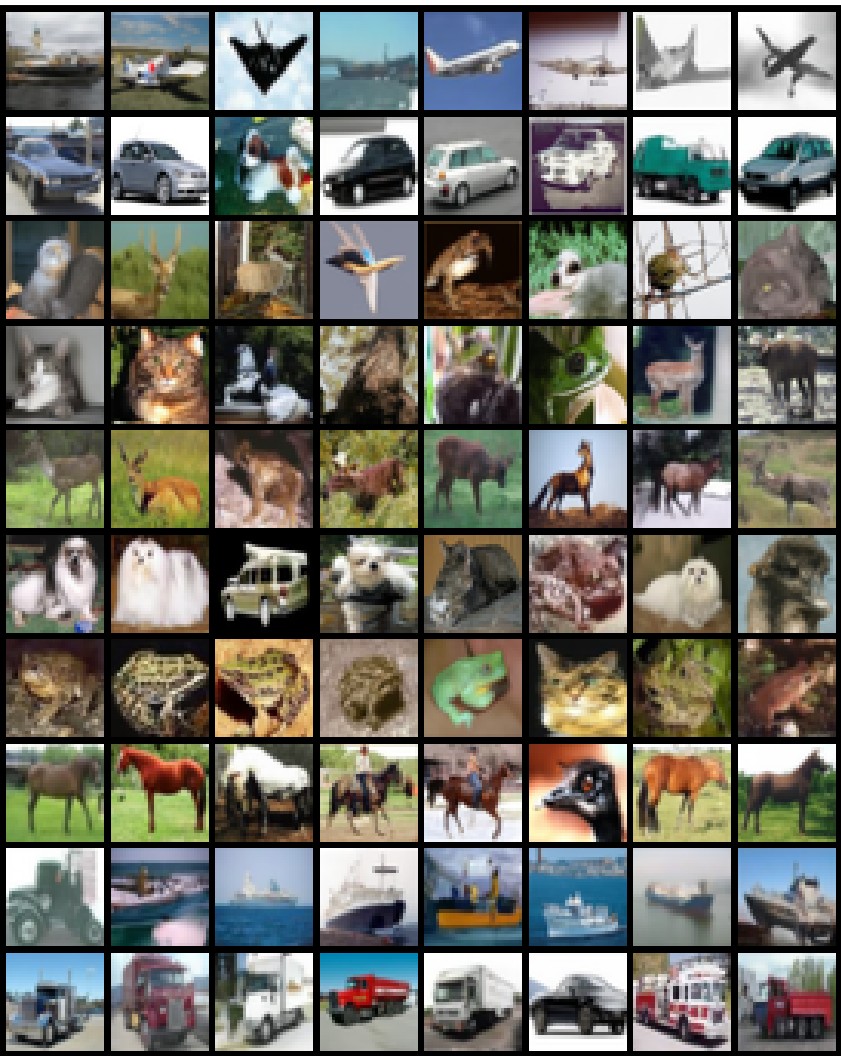

Figure 15: Randomly selected images of vanilla classifier guidance (40% labeled data)

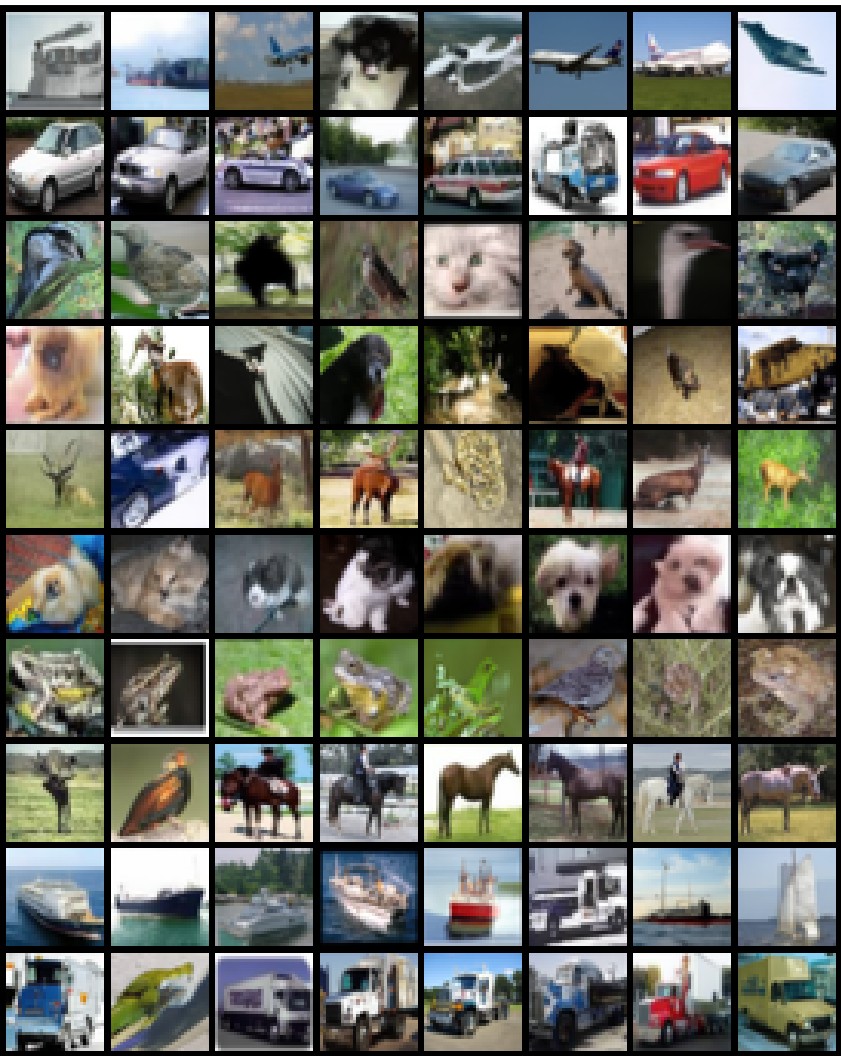

Figure 16: Randomly selected images of vanilla classifier guidance (60% labeled data)

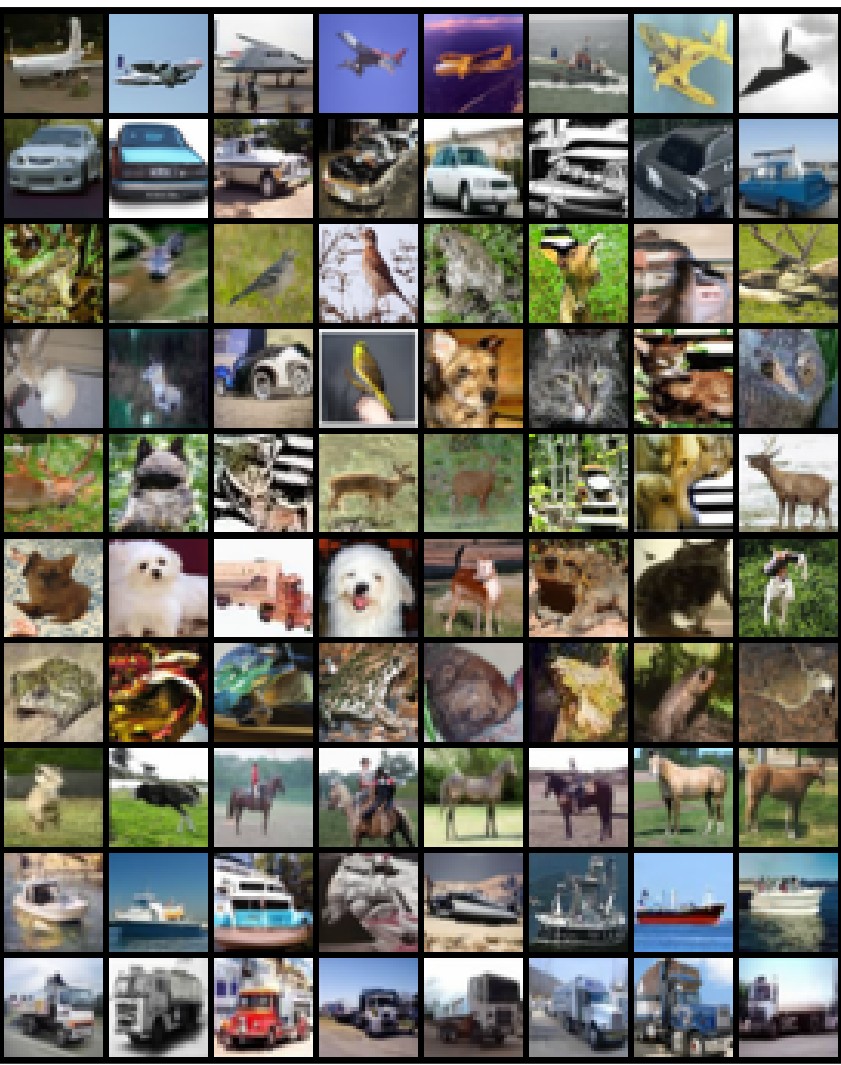

Figure 17: Randomly selected images of vanilla classifier guidance (80% labeled data)

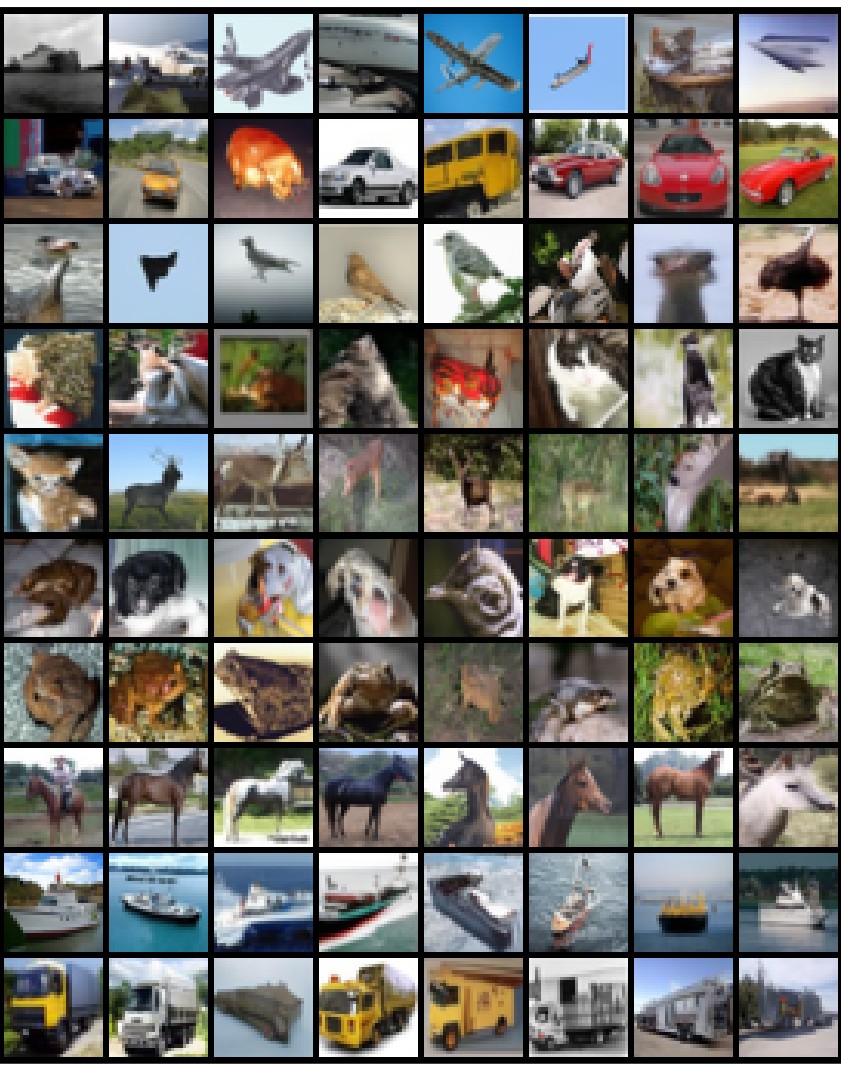

Figure 18: Randomly selected images of vanilla classifier guidance (100% labeled data)

