# OpenReview forum: "Score-based Conditional Generation with Fewer Labeled Data by Self-calibrating Classifier Guidance"
_ICLR.cc/2024/Conference — Submitted to ICLR 2024_

### Official Review · Reviewer_yr3t · 2023-10-15

**Soundness:** 2 fair
**Presentation:** 3 good
**Contribution:** 3 good
**Rating:** 5
**Confidence:** 5

**Summary:**

This paper introduces a regularization approach for improving the training of Classifier-Guided Score-based Generative Models (CGSGM). The proposed method leverages the interpretation of the Joint Energy-based Model (JEM) model and adopts the Denoising Score Matching (DSM) loss in the optimization process to enhance its performance. The experiments show that the proposed Self-Calibration (SC) regularization loss is effective in conditional generation in terms of the intra-FID metric under the context of semi-supervised learning.

**Strengths:**

- Considering the computational cost, the proposed SC loss seems to be a more appealing option in comparison to the JEM’s maximum likelihood objective in augmenting the classifiers' ability to capture the underlying energy function.
- The comprehensive review in Section 2 offers readers a holistic perspective, aiding in their understanding of the background information related to CGSGM. In addition, the illustrative example in Fig. 3 serves as empirical evidence supporting the effectiveness of the proposed SC loss.
- The proposed method has demonstrated enhanced performance compared to other baseline methods on the selected two datasets when working with limited accessible labels.

**Weaknesses:**

- The current formulation of the proposed method appears to lack conciseness, given that both $s(\boldsymbol{x},t;\phi)$ and $\nabla_\boldsymbol{x} \log \sum_{y} f(\boldsymbol{x}, y,t;\phi)$ can serve as parameterized score functions. In the context of the CG framework, one of these components should be redundant.
- Table 1 lacks a performance comparison between the introduced method and the JEM framework. Given that the design of the proposed method bears considerable resemblance to JEM, it seems necessary to have a performance evaluation against JEM on real-world datasets. To incorporate the diffusion process into the JEM framework (i.e., the CG-JEM setup mentioned in Section 3.2) in real-world setups, the reviewer suggests using the method presented in [1].
- Diffusion score-based models have exhibited state-of-the-art performance for years, showcasing exemplary results on numerous benchmarks with photo-realistic data. However, the experiments in this paper are confined to an initial setup, focusing on images with a 32x32 resolution and only encompassing 100 classes. The reviewer is uncertain whether the proposed approach can demonstrate superior performance among the baselines on widely recognized benchmarks such as ImageNet.

**Questions:**

- The rationale behind the design of the proposed SC loss remains ambiguous. Could the authors elucidate with an analytical perspective why the proposed method should outperform the baselines (i.e., DLSM and JEM), which explicitly minimize the Fisher and KL divergences, respectively? Furthermore, could the authors theoretically justify the effectiveness of the proposed loss in a semi-supervised setup (e.g., defining the relationship of data-label pairs through optimal transport [2])?
- How were the hyper-parameters for the baseline methods, specifically the balancing factors of CG-DLSM and CG-JR, and the smoothing factor in CG-LS, chosen to ensure a fair comparison in Section 4?
____
Overall, I believe that semi-supervised CGSGM holds the potential to become a promising research direction. However, because of the aforementioned concerns and questions, I am giving this paper a borderline score. I am open to revising the score upwards if the issues and queries are adequately addressed.
____
**References**

[1] Gao *et al.* Learning Energy-Based Models by Diffusion Recovery Likelihood, *ICLR*, 2021.\
[2] Seguy *et al.* Large-Scale Optimal Transport and Mapping Estimation, *ICLR*, 2018.

---

> ### Author Response · Authors · 2023-11-22
> **Reply to Reviewer yr3t by Authors**
>
> We appreciate your thoughtful feedback on our paper. We have carefully addressed every concern of yours, as discussed below.
> - [Weakness 1] One of the two score estimators in the proposed framework is redundant
>     - We agree that the existence of two score estimators is somewhat redundant. The redundancy that resulted from the auxiliary task of score matching, however, should not be considered a burden to the design. Other studies [1] that introduce auxiliary tasks to deep learning models are considered effective even though the extra task is not performed during testing time and is thus redundant. The auxiliary task introduced in our work should be considered beneficial since the performance of the main task is improved significantly.
> - [Weakness 2] Compare with JEM using "Learning Energy-Based Models by Diffusion Recovery Likelihood" (Gao et al, 2021)
>     - Thank you for the suggestion. We have started running experiments using this method after we received the review. However, since it is really slow to train EBMs as MCMC sampling is needed, we only have some preliminary results by now (i.e. images generated using early checkpoints). We will include complete results in future revisions when the experiments are finished. Here are the preliminary results for CG-JEM using the suggested method:
>
>         |Dataset|% Labeled data|intra-FID|Acc|FID|IS|
>         |-|-|-|-|-|-|
>         |CIFAR-10|5%|25.91|0.519|2.803|10.00|
>         |CIFAR-10|20%|21.02|0.590|2.75|10.10|
>         |CIFAR-10|100%|28.78|0.451|2.13|9.87|
>
>         |Dataset|% Labeled data|intra-FID|FID|IS|
>         |-|-|-|-|-|
>         |CIFAR-100|5%|121.97|4.33|12.11|
>         |CIFAR-100|20%|104.02|3.74|12.47|
>         |CIFAR-100|100%|93.99|3.68|11.86|
> - [Weakness 3] The method is only evaluated on datasets with 32x32 resolution and $\leq 100$ classes (CIFAR-10 and CIFAR-100)
>     - We agree on the importance of evaluating our method on more datasets with higher resolution and more classes. Due to the limited computational resources in the academic environment, we were not able to do so previously. We are continuously using our available resources for additional experiments and will include them in future revisions upon completion. In the meantime, we hope to emphasize that the clear advantage of our proposed CG-SC in the fewer labeled data regime in CIFAR-10 and CIFAR-100 already provides strong support for the validity of the proposed method.
> - [Question 1] Elucidate with an analytical perspective why the proposed method should outperform the baselines (JEM, DLSM)?
>     - Advantage over JEM:
>     	- Time efficiency: Since the training of JEM requires MCMC sampling, further interpreting classifiers as SGMs significantly reduces training time.
> 	- Advantage over DLSM:
> 		- Utilization of unlabeled data: The calculation of DLSM loss requires ground truth labels, while the proposed loss can be applied to both labeled and unlabeled data.
> - [Question 2] Could the authors theoretically justify the effectiveness of the proposed loss in a semi-supervised setup (e.g., defining the relationship of data-label pairs through optimal transport)? (Seguy et al. Large-Scale Optimal Transport and Mapping Estimation, ICLR, 2018)
>     - To our understanding, the theoretical effectiveness of our method is not backed by optimal transport. We agree that given the promising results, further theoretical justifications would be beneficial. We hope our work could stimulate future theoretical analysis of the effect of the proposed regularization method.
> - [Question 3] How were the hyper-parameters for the baseline methods, specifically the balancing factors of CG-DLSM and CG-JR, and the smoothing factor in CG-LS, chosen to ensure a fair comparison in Section 4?
> 	- Balancing factor of CG-DLSM is selected to be $1$, as suggested in the original paper.
> 	- Balancing factor of CG-JR is selected to be $0.01$, as suggested in the original paper.
> 	- The smoothing factor in CG-LS is tuned between $\{0.1, 0.05\}$ for the better intra-FID. Note that the performance of the two settings do not differ by much.
>
> [1] Liebel et al. Auxiliary Tasks in Multi-task Learning, arXiv, 2018

---

### Official Review · Reviewer_4JFE · 2023-10-31

**Soundness:** 3 good
**Presentation:** 3 good
**Contribution:** 2 fair
**Rating:** 5
**Confidence:** 3

**Summary:**

The paper focuses on the development of a classifier guidance score-based generative model (CGSGM) for the semi-supervised setting with limited labeled data. The authors introduce a self-calibration (SC) loss that internally calibrates the classifier without depending on the unconditional SGM. Drawing inspiration from the reinterpretation of classifiers as energy-based models (EBM) as proposed by JEM, the authors devise a loss function trainable with both labeled and unlabeled data. This approach not only bolsters the generative capability of the classifier but also enhances the accuracy of the learned conditional distribution scores, which are computed from the classifier's logits output, thereby augmenting its conditional generative capacity. In terms of experimental validation, the authors initially test on a toy dataset using SC loss, illustrating how the SC loss improves classifiers by generating precise conditional distribution scores. They then contrast the CGSGM-SC approach with the existing vanilla classifier-guidance method (CG), various regularization methods designed to refine the classifier (CG-DLSM, CG-LS, CG-JR), and the classifier-free method (CFG) on the CIFAR-10 and CIFAR-100 datasets. The results demonstrate the superiority of their method in a semi-supervised setting with sparse labeled data.

**Strengths:**

1. This paper is commendable for its clear articulation of motivation. The writing exhibits a logical structure, and the methodology employed is both concise and straightforward, making it easy for readers to understand.
2. This paper has conducted extensive and thorough experiments comparing with all existing conditional score-based generative models, demonstrating the enhanced image generation capability of the classifier trained with the proposed SC loss (as shown in the Appendix G) and its superior performance in generation in the semisupervised setting with fewer labeled data (Sec 4.2).

**Weaknesses:**

1. This work is the lack of technical novelty. The proposed self-calibration (SC) loss in this paper heavily depends on two prior works. One is EBMs, which connect probability distribution and energy function. The other one is JEM, which reinterprets classifiers $f(\boldsymbol{x}, y ; \boldsymbol{\phi})$ as EBMs and enforcing regularization with related objectives helps classifiers $f(\boldsymbol{x}, y ; \boldsymbol{\phi})$ to capture more accurate probability distributions.

2. While the method proposed in this paper demonstrates stronger capabilities when labeled data is extremely scarce (only 5% of the data is labeled), in many instances where labeled data exceeds 50% (as shown in Sec 4.2 and Appendix A), the CFG method exhibits greater conditional generative power than the proposed CG-SC method and traditional CG methods. Currently, the CFG method is the more prevalent approach in conditional generation, mainly because it eliminates the need for training an additional classifier and can seamlessly integrate into the conventional DSM loss, which is much more easier to be applied in downstream tasks.

**Questions:**

1. What is the hyperparameter $\lambda_{CG}$ for classifier guidance with and without self-calibration set in Table 1 for CG and CG-SC methods? As the traditional CG method regularizes the classifier externally by $\lambda_{CG}$ and the choice of the parameter $\lambda_{CG}$ chosen has a significant impact on the generation outcomes[1], there may be more experiment for CG by tuning the hyperparameter $\lambda_{CG}$ for a fair comparison. Moreover, there is also a hyperparameter classifier-free guidance scale $\lambda_{CFG}$ for CFG method [2], there should be more experiment on CFG methods by tuning the hyperparameter $\lambda_{CFG}$ for a fair comparison.

2. Why there is an inconsistency between conditional metrics and unconditional metrics? (Appendix H) The FID (Fréchet Inception Distance) serves as a metric to assess the diversity and authenticity of generated data distributions in comparison to true data distributions. In my view, if the conditional metrics for each category (intra FID) are better, then the overall unconditional metrics (FID) should also be better.

[1] Dhariwal P, Nichol A. Diffusion models beat gans on image synthesis[J]
[2] Ho J, Salimans T. Classifier-free diffusion guidance[J]. arXiv preprint arXiv:2207.12598, 2022

---

> ### Author Response · Authors · 2023-11-22
> **Reply to Reviewer 4JFE by Authors**
>
> We appreciate your thoughtful feedback on our paper. We have carefully addressed every concern of yours, as discussed below.
> - [Weakness 1] Lack of novelty: depends on JEM and EBM
>     - Although the proposed method utilizes the findings of JEM and EBM, we believe it is unfair to judge our novelty based on this. Our main contributions are that we proposed to further reinterpret classifiers as SGMs for regularization purposes, discovered the potential of CGSGM under semi-supervised settings, and designed experiments to verify the proposed method's effectiveness in improving CGSGM under both fully-supervised and semi-supervised settings.
> - [Weakness 2] In instances where labeled data exceeds 50% (as shown in Sec 4.2 and Appendix A), the CFG method exhibits greater conditional generative power than the proposed CG-SC method and traditional CG methods.
>     - We agree about the strength of CFG (and Cond) with more labeled data. Somehow we hope to re-emphasize that the paper does not focus on the regime of more labeled data. We believe it would be fairer if our proposed CG-SC could be discussed within the regime that we designed the method for. The results demonstrate CG-SC is more effective in the more realistic or common scenarios with less than 20% labeled data---the focus of our paper.
> - [Question 1] Tuning $\lambda_{CG}$ and $\lambda_{CFG}$
>     - Both parameters are tuned to obtain the best intra-FID
>     - $\lambda_{CG}$ is tuned within $\{0.5, 0.8, 1.0, 1.2, 1.5, 2.0, 2.5\}$
>     - $\lambda_{CFG}$ is tuned within $\{0, 0.1, 0.2, 0.4\}$
>     - We will include this in the Appendix of future revisions for further clarification of the experimental details.
> - [Question 2] Why is there a mismatch between conditional and unconditional metrics?
>     - Although the generated images exhibit high fidelity and cover a significant portion of the input data distribution (good FID), they might fail to accurately capture the class-conditional distributions (poor intra-FID). This limitation can result from either poor generation accuracy or low generation diversity in certain classes, leading to poor intra-FID scores for those specific classes, as shown in Appendix A. For instance, consider an extreme case where the generator mistakenly swapped images between the 'cat' and 'dog' classes, the FID may remain high, but the intra-FID would significantly decline. After we swap the incorrect images back, the intra-FID would become substantially better while the FID remains the same.
>     - Tuning the scaling factor introduces a trade-off between accuracy and diversity. In Appendix H, as the scaling factor becomes larger, the generation accuracy increases while the diversity decreases. The mismatch between FID and intra-FID came from the fact that FID only considers the diversity of images, while intra-FID additionally takes into account how well the generated images represent each class (generation accuracy).

---

> ### Comment · Area_Chair_eJZu · 2023-12-04
> **[Important] Response Required to Authors' Rebuttal**
>
> Dear Reviewer 4JFE,
>
> As we progress through the review process for ICLR 2024, I would like to remind you of the importance of the rebuttal phase. The authors have submitted their rebuttals, and it is now imperative for you to engage in this critical aspect of the review process.
>
> Please ensure that you read the authors' responses carefully and provide a thoughtful and constructive follow-up. Your feedback is not only essential for the decision-making process but also invaluable for the authors.
>
> Thank you,
>
> ICLR 2024 Area Chair

---

### Official Review · Reviewer_zxWa · 2023-11-01

**Soundness:** 3 good
**Presentation:** 3 good
**Contribution:** 3 good
**Rating:** 5
**Confidence:** 3

**Summary:**

The paper argues that the performance of classifier guidance score generative models(CGSGM) under limited labeled data setting is bottlenecked by over-fitting of the classifier whose conditional scores may be unreliable. The paper argues that while there are several regularization methods to prevent over-fitting of the classifiers the score of such classifiers is often not aligned with the unconditional SGM’s view of the underlying distribution and thus offers limited benefits. While other approaches exist to solve this issue of distribution alignment, they regularize the classifier by using an external SGM. In contrast, the paper does not use any external SGM to regularize the classifier, by taking inspiration from Joint energy-based models which interpret classifiers as energy-based models. The authors propose a self-calibration loss for regularizing the classifier without using using any external SGM and thus can train the classifier and the SGM on both labeled and unlabeled datasets.

With experiments on CIFAR-10 and CIFAR-100 on only 5% of the labeled data the authors show improved performance in intra-FID and accuracy metrics compared to other CGSGM approaches.

**Strengths:**

1. The experimental results for CIFAR-10 and CIFAR-100 are good an show the efficacy of the proposed approach.
2. The theoretical analysis done by the paper is sound.

**Weaknesses:**

1. While the paper mentions that simply regularizing the classifier to prevent over-fitting suffers from distribution alignment, not much experiments are done to justify this. The only baseline which the paper considers is the adversarial training of the classifier. I would like to see more experiments done with strong regularizers such as using RandAugment or CutMix augmentations.

2. Further what would happen if the classifier is itself trained in a semi-supervised fashion with methods such as [1]. Such semi-supervised approaches perform very good with a limited data. The authors should do experiments with classifiers trained with such semi-supervised approaches.

3. For the CIFAR-100 results in table1 why doesn't it have the accuracy metric?

4. For the baseline CG-DLSM, the classifier is regularized over only labeled data using external SGM. However, this classifier can be regularized over unlabeled data also? What happens if this baseline's classifier is also regularized over unlabeled data? This would be a fairer comparison with the paper's results.

References -
1. FixMatch: Simplifying Semi-Supervised Learning with Consistency and Confidence. Sohn et al. https://arxiv.org/pdf/2001.07685.pdf

**Questions:**

I have already mentioned it in the weakness section

---

> ### Author Response · Authors · 2023-11-22
> **Reply to Reviewer zxWa by Authors**
>
> We appreciate your thoughtful feedback on our paper. We have carefully addressed every concern of yours, as discussed below.
> - [Weakness 1] Experiments with RandAugment and CutMix
>     - Indeed augmentation-based regularizations are important. We wanted to avoid augmentation-based regularization since it is hard to compare those methods with regularizations that use auxiliary losses as they are done in different stages (pre-processing v.s. training stage).
> - [Weakness 2] Experiments with semi-supervised approaches
>     - In the current stage, we've tried incorporating Fixmatch [1] and EMA-teacher [2] to train the time-dependent classifier. While it leads to some improvement over vanilla CGSGM, it often leads to collapsed conditional generation results of some classes. For example, the images generated with the label "airplane" might contain images generated from random classes while the generative performance of other classes improves. Current results suggest that integrating semi-supervised methods into time-dependent classification is non-trivial and requires further exploration.
> - [Weakness 3] Why doesn't Cifar-100 have generation accuracy evaluation?
>     - As mentioned in Section 4.1, nearly-accurate classification is desired for meaningful evaluation of the generation accuracy, but we were unable to locate such a classifier for the CIFAR-100 dataset. Therefore, generation accuracy is not included for the CIFAR-100 dataset.
> - [Weakness 4] Why not use DLSM on unlabeled data?
>     - In the formulation of DLSM loss $\mathbb{E}\left[\frac{1}{2}\lVert \nabla_\tilde{x}\log p(\tilde{y}\vert \tilde{x}; \theta)+s(\tilde{x};\phi)-\nabla_\tilde{x}\log p_\sigma(\tilde{x}\vert x)\rVert^2\right]$, ground truth labels are required for calculation (the $\tilde{y}$ in $\nabla_\tilde{x}\log p(\tilde{y}\vert \tilde{x}; \theta)$). As a result, it is not applicable to unlabeled data.
>
> [1] Sohn et al. FixMatch: Simplifying Semi-Supervised Learning with Consistency and Confidence, NeurIPS, 2020
>
> [2] Cai et al. Exponential Moving Average Normalization for Self-supervised and Semi-supervised Learning, CVPR, 2021

---

> ### Comment · Area_Chair_eJZu · 2023-12-04
> **[Important] Response Required to Authors' Rebuttal**
>
> Dear Reviewer zxWa,
>
> As we progress through the review process for ICLR 2024, I would like to remind you of the importance of the rebuttal phase. The authors have submitted their rebuttals, and it is now imperative for you to engage in this critical aspect of the review process.
>
> Please ensure that you read the authors' responses carefully and provide a thoughtful and constructive follow-up. Your feedback is not only essential for the decision-making process but also invaluable for the authors.
>
> Thank you,
>
> ICLR 2024 Area Chair

---

### Official Review · Reviewer_BW7Z · 2023-11-01

**Soundness:** 3 good
**Presentation:** 3 good
**Contribution:** 3 good
**Rating:** 5
**Confidence:** 4

**Summary:**

This methodology innovatively reinterprets classifiers as time-dependent Energy-Based Models, offering enhanced regularization through a gradient-based approach and introducing a novel self-calibration loss. These advances significantly improve alignment with underlying data distributions, boosting generalization and robustness in classifier-guided generative modeling.

The approach enables more effective utilization of unlabeled data instances by integrating them into the learning process through self-calibration, enhancing the model's ability to capture and represent the broader, underlying data distribution.

**Strengths:**

This methodology excels in intra-FID scoring, which is crucial for class-wise interpretation, particularly in scenarios with partially labeled data, demonstrating its effectiveness in maintaining class-specific fidelity even with limited label availability.

The approach of using score-based conditional generation in a semi-supervised learning context is not a commonly explored direction in research. This lends a novel aspect to the problem being addressed by this methodology.

**Weaknesses:**

The methodology appears incremental, building marginally upon JEM's foundation of interpreting classifiers as time-dependent EBMs. The newly introduced self-calibration loss primarily enhances this by applying a standard DSM technique to train the internal score function, thus lacking substantial novelty.

The authors have judiciously selected a range of baseline candidates for semi-supervised learning and reported performance results. However, the focus on datasets like CIFAR-10 and CIFAR-100 limits the assessment of the methodology's generalizability to more diverse or complex data scenarios.

* Minor weaknesses
The allocation of Figure 1 is too naive. Overall, you could have edited the space of main paper more wisely.

**Questions:**

Q1. Does the author's argument also suggest that, compared to Deep Generative Models (DGMs), classifier architectures may be more prone to over-fitting?

Q2. Why are the FID (Fréchet Inception Distance) scores for CFG-all or CFG-labeled methodologies notably poor?

---

> ### Author Response · Authors · 2023-11-22
> **Reply to Reviewer BW7Z by Authors**
>
> We appreciate your thoughtful feedback on our paper. We have carefully addressed every concern of yours, as discussed below.
> - [Weakness 1] Our method is incremental (adding JEM and DSM to train the classifier)
>     - Although the proposed method utilizes the findings of JEM and DSM, we believe it is unfair to judge our novelty based on this. Our main contributions are that we proposed to further reinterpret classifiers as SGMs for regularization purposes, discovered the potential of CGSGM under semi-supervised settings, and designed experiments to verify the proposed method's effectiveness in improving CGSGM under both fully-supervised and semi-supervised scenarios.
> - [Weakness 2] The focus on datasets like CIFAR-10 and CIFAR-100 limits the assessment of the methodology's generalizability to more diverse or complex data scenarios.
>     - We agree on the importance of evaluating our method on more datasets with higher resolution and more classes. Due to the limited computational resources in the academic environment, we were not able to do so previously. We are continuously using our available resources for additional experiments and will include them in future revisions upon completion. In the meantime, we hope to emphasize that the clear advantage of our proposed CG-SC in the fewer labeled data regime in CIFAR-10 and CIFAR-100 already provides strong support for the validity of the proposed method.
> - Minor weaknesses
>     - Thank you for the suggestions. We will improve the presentation of the mentioned point in future revisions.
> - [Question 1] Does the author's argument also suggest that, compared to Deep Generative Models (DGMs), classifier architectures may be more prone to over-fitting?
>     - Both classifiers and deep generative models have over-fitting problems, and it is hard to determine which is more prone to over-fitting. What we want to say is that over-fitting classifiers are harmful to the generative performance of CGSGM. Therefore, the classifier needs to be regularized for better generative performance.
> - [Question 2] Why are FID in CFG-all and CFG-labeled notably poor?
>     - As discussed in Section 4.2, this is because those methods generate samples with degraded diversity under semi-supervised scenarios. We observed that CFG-all and CFG-labeled tend to generate samples highly correlated to the labeled data. The lack of labeled data causes those methods to generate images with far less diversity, making their performance much worse when evaluated using FID and intra-FID.

---

> > ### Comment · Reviewer_BW7Z · 2023-11-23
> > **Thanks for the author's response**
> >
> > Thank you for your response. While some aspects have become clearer, I believe it's hard to fully assess the efficacy of the methodology without experimental results on datasets with higher resolution and more classes. Therefore, I intend to maintain my current score.

---

### Official Review · Reviewer_b4HY · 2023-11-09

**Soundness:** 2 fair
**Presentation:** 3 good
**Contribution:** 2 fair
**Rating:** 5
**Confidence:** 3

**Summary:**

This paper aims to solve the problem of conditional generation with fewer labeled data. The authors propose a new method to improve classifier-guided score-based generation by regularizing the classifier during training time. The key idea of this regularizer is to use principles from energy-based models to convert the classifier as another view of the unconditional SGM. Experimental results illustrate the superiority of the proposed method over several related baselines.

**Strengths:**

First, I am not an expert in conditional score-based generation and may miss some related work.

## Quality
* Experimental results illustrate the superiority of the proposed method.

## Clarity
* Overall, this paper is very well-written.
* The related works are discussed clearly.

## Significance
* The paper can contribute to the area of semi-supervised conditional generalization.

**Weaknesses:**

## Originality
* I am afraid that the novelty may be limited since the techniques are common in deep-generation models.

## Quality & Clarity
* For the proposed method, the classification ability of the trained classifier is not tested experimentally.

## Significance
* The proposed method may have little effect on other sub-fields of machine learning.

**Questions:**

1. In my view, the proposed method involves a classifier, so why not test its classification ability?
2. Please discuss the relationship with a recent highly relevant paper [1*].

[1*] Diffusion Models and Semi-Supervised Learners Benefit Mutually with Few Labels, NeurIPS 2023

---

> ### Author Response · Authors · 2023-11-22
> **Reply to Reviewer b4HY by Authors**
>
> We appreciate your thoughtful feedback on our paper. We have carefully addressed every concern of yours, as discussed below.
> - [Weakness 1] Originality: The techniques are common in deep-generation models
>     - Indeed, regularization methods are common in deep generative models, but whether a new regularization technique is effective needs verification through experiments. Our main contributions are that we proposed to further reinterpret classifiers as SGMs for regularization purposes, discovered the potential of CGSGM under semi-supervised settings, and designed experiments to verify the proposed method's effectiveness in improving CGSGM under both fully-supervised and semi-supervised scenarios. Furthermore, to our understanding, there is no other work that proposed to regularize classifiers as SGMs.
> - [Weakness 2 & Question 1] Why didn't the authors test the classification accuracy of the classifiers?
>     - The main goal of our study is to improve the generative performance of CGSGMs instead of improving classification or joint modeling. Related works on CGSGMs [1,2,3] also reported only generative performance.
>     - Nevertheless, we did evaluate the classification accuracy (at timestep $t=0$) of CG and CG-SC-all for the CIFAR-10 dataset in the preliminary stage of our work. Despite observing improvement over CG, the classification accuracies for both methods are poor compared to the SOTA of ordinary classification for the CIFAR-10 dataset. Note that this is similar to the observation reported [here](https://openreview.net/forum?id=AAWuCvzaVt#:~:text=We%20have%20not,traditional%20ImageNet%20classifiers.) by authors of [1]. It is hard to tell whether these classifiers perform well, given that the time-dependent semi-supervised classification task is substantially more challenging than ordinary semi-supervised classification.
> 		|% labeled data|CG|CG-SC|
> 		|-|-|-|
> 		|100%|0.808|0.925|
> 		|80%|0.803|0.909|
> 		|60%|0.752|0.905|
> 		|40%|0.680|0.731|
> 		|20%|0.571|0.738|
> 		|5%|0.420|0.658|
> 	- Although improvement was observed through evaluating classification accuracy, we could not explicitly explain how well the time-dependent classifiers are performing, and how testing accuracies of such classifiers affect the generation quality of CGSGMs is still to be examined. We aimed to provide evaluation metrics that more directly reflect the generation performance of different methods. As a result, we chose to present the current evaluation metrics (FID, intra-FID, inception score, and generation accuracy) in our work.
> - [Weakness 3] Little effect on other sub-fields of machine learning
>     - We believe our work should already be considered significant as the sub-field we are studying ( conditional generation) is itself a significant sub-field of machine learning.
> - [Question 2] Discuss the relationship with [4]
>     - Their work focuses on improving the performance of **ordinary** semi-supervised classifiers and conditional diffusion models using dual training.
>     - Our work focuses on improving classifier-guided SGMs through regularizing the **time-dependent** classifier using both labeled and unlabeled data.
>
> Link 1: https://openreview.net/forum?id=AAWuCvzaVt#:~:text=We%20have%20not,traditional%20ImageNet%20classifiers.
>
> [1] Dhariwal et al. Diffusion Models Beat GANs on Image Synthesis, NeurIPS, 2021
>
> [2] Chao et al. Denoising Likelihood Score Matching for Conditional Score-based Data Generation, ICLR, 2022
>
> [3] Kawar et al. Enhancing Diffusion-Based Image Synthesis with Robust Classifier Guidance, TMLR, 2023
>
> [4] Diffusion Models and Semi-Supervised Learners Benefit Mutually with Few Labels, NeurIPS 2023

---

> ### Comment · Area_Chair_eJZu · 2023-12-04
> **[Important] Response Required to Authors' Rebuttal**
>
> Dear Reviewer b4HY,
>
> As we progress through the review process for ICLR 2024, I would like to remind you of the importance of the rebuttal phase. The authors have submitted their rebuttals, and it is now imperative for you to engage in this critical aspect of the review process.
>
> Please ensure that you read the authors' responses carefully and provide a thoughtful and constructive follow-up. Your feedback is not only essential for the decision-making process but also invaluable for the authors.
>
> Thank you,
>
> ICLR 2024 Area Chair

---

### Author Response · Authors · 2023-11-22
**Summary of Rebuttal Revision**

We highly appreciate the questions and suggestions from all reviewers. We have revised some parts of our paper to address some of the concerns. The revision can be summarized as the following:
- Adjusted the positioning of Figure 1
- Some updates in the introduction
- Include a summary of our contribution at the end of introduction
- Include more experimental details in Appendix D

---

### Meta-Review · Area_Chair_eJZu · 2023-12-09

**Metareview:**

The paper proposes a novel methodology incorporating self-calibration (SC) loss for augmenting classifiers in conditional score-based generative models (CGSGM), aiming to enhance performance in semi-supervised learning contexts. Despite an average review score of 5, suggesting a borderline situation, the recommendation leans toward rejection. The reviewers acknowledge the methodology's strengths, including its originality in combining SC loss with CGSGM and its effectiveness in certain experimental settings. However, several significant weaknesses have been identified, which the authors have not fully addressed.

**Justification For Why Not Higher Score:**

Considering the identified weaknesses and the authors' partial response to these concerns, the recommendation is inclined toward rejection. The paper, while demonstrating some promising aspects, falls short in terms of technical novelty, analysis, and the breadth of experimentation. The concerns about the paper's adequacy of its theoretical foundations and the fairness and thoroughness of its experimental evaluations suggest further improvements for the paper.

**Justification For Why Not Lower Score:**

N/A

---

### Decision · Program_Chairs · 2024-01-16

Reject